# ON THE ROLE OF MOMENTUM IN THE IMPLICIT BIAS OF GRADIENT DESCENT FOR DIAGONAL LINEAR NETWORKS

## ABSTRACT

Momentum is a widely adopted and crucial modification to gradient descent when training modern deep neural networks. In this paper, we target on the regularization effect of momentum-based methods in regression settings and analyze the popular diagonal linear networks to precisely characterize the implicit bias of continuous version of heavy-ball (HB) and Nesterov's method of accelerated gradients (NAG). We show that, HB and NAG exhibit different implicit bias compared to GD for diagonal linear networks, which is different from the one for classic linear regression problem where momentum-based methods share the same implicit bias with GD. Specifically, the role of momentum in the implicit bias of GD is twofold. On one hand, HB and NAG induce extra initialization mitigation effects similar to SGD that are beneficial for generalization of sparse regression. On the other hand, besides the initialization of parameters, the implicit regularization effects of HB and NAG also depend on the initialization of gradients explicitly, which may not be benign for generalization. As a consequence, whether HB and NAG have better generalization properties than GD jointly depends on the aforementioned twofold effects determined by various parameters such as learning rate, momentum factor, data matrix, and integral of gradients. Particularly, the difference between the implicit bias of GD and that of HB and NAG disappears for small learning rate. Our findings highlight the potential beneficial role of momentum and can help understand its advantages in practice from the perspective of generalization.

## 1 INTRODUCTION

Extensive deep learning tasks aim to solve the optimization problem

$$\arg\min_{\beta} L(\beta) \tag{1}$$

where $L$ is the loss function. Gradient descent (GD) and its variants underpin such optimization of parameters for deep learning, thus understanding these simple yet highly effective algorithms is crucial to unveil the thrilling generalization performance of deep neural networks. Recently, Azulay et al. (2021); Ji & Telgarsky (2019); Lyu & Li (2020); Soudry et al. (2018); Pesme et al. (2021) have made significant efforts in this direction to understand GD-based methods through the lens of *implicit bias*, which states that *GD and its variants are implicitly biased towards selecting particular solutions among all global minimum.*

In particular, Soudry et al. (2018) pioneered the study of implicit bias of GD and showed that GD selects the max-margin solution for logistic regression on separable dataset. For regression problems, the simplest setting is the linear regression problem, where GD and its stochastic variant, SGD, are biased towards the interpolation solution that is closest to the initialization measured by the Euclidean distance (Ali et al., 2020). In order to investigate the implicit bias for deep neural networks, diagonal linear network, a simplified version of deep learning models, has been proposed. For this model, Azulay et al. (2021); Woodworth et al. (2020); Yun et al. (2021) showed that the solution selected by GD is equivalent to that of a constrained norm minimization problem interpolating between $\ell_1$ and $\ell_2$ norms up to the magnitude of the initialization scale. Pesme et al. (2021) further characterized that adding stochastic noise to GD additionally induces a regularization effect equivalent to reducing the initialization magnitude.

Besides these fruitful progresses, Gunasekar et al. (2018); Wang et al. (2022) studied the implicit bias of the widely adopted modification to GD, the momentum-based methods for one-layer linear models and showed that they have the same implicit bias as GD. Jelassi & Li (2022), on the other hand, revealed that momentum-based methods have better generalization performance than GD for a linear CNN in classification problems. Ghosh et al. (2023) conducted a model-agnostic analysis of $\mathcal{O}(\eta^2)$ approximate continuous version of HB and revealed its generalization advantages over GF from the perspective of IGR (Barrett & Dherin, 2022). Momentum induces different dynamics compared to vanilla GD: from the perspective of their continuous time approximation modelling, the approximation for GD is a first-order ODE (gradient flow): $d\beta/dt = -\nabla L(\beta)$, while, as a comparison, the approximation for momentum-based methods can be seen as a damped second-order Hamiltonian dynamic with potential $L(\beta)$:

$$m\frac{d^2\beta}{dt^2} + \lambda\frac{d\beta}{dt} + \nabla L(\beta) = 0,$$

which was first inspired in Polyak (1964). Due to the clear discrepancy of their dynamics, it is natural and intriguing to ask from the theoretical point of view:

**(Q):** *Will adding GD with momentum change its implicit bias for deep neural networks?*

For the least squares problem (single layer linear network), Gunasekar et al. (2018) argued that momentum does not change the implicit bias of GD, while it is unclear if this is the case for deep learning models. However, from empirical point of view, momentum is a crucial and even inevitable component in the training of modern deep neural networks practically. Thus its theoretical characterization is significant and necessary, especially considering that momentum-based methods typically lead to a better generalization performance which suggests that they might enjoy a different implicit bias compared to GD.

Hence, our goal in this work is to precisely characterize the implicit bias of momentum-based methods to take the first step towards answering the above fundamental question. To explore the case for deep neural works, we consider the popular deep linear models: *diagonal linear networks*. Although the structures of diagonal linear networks are simple, they already capture many insightful properties of deep neural networks, including the dependence on the initialization, the over-parameterization of the parameters and the transition from lazy regime to rich regime (Woodworth et al., 2020; Pesme et al., 2021), an intriguing phenomenon observed in many complex neural networks.

**Our contribution.** Heavy-Ball algorithms (Polyak, 1964) (HB) and Nesterov's method of accelerated gradients (Nesterov, 1983) (NAG) are the most widely adopted momentum-based methods. These algorithms are generally implemented with a fixed momentum factor in deep learning libraries such as PyTorch (Paszke et al., 2017). To be consistent with such practice, we focus on HB and NAG with a fixed momentum factor that is independent of learning rate or iteration count. For the purpose of conducting a tractable theoretical analysis, we rely on the tools of continuous time approximations of momentum-based methods (with fixed momentum factor), HB and NAG flow, which were recently interpreted by Kovachki & Stuart (2021) as *modified equations* in the numerical analysis literature and by Shi et al. (2018) as *high resolution ODE* approximation. Our findings are summarized below.

We show that, *unlike* the case for single layer linear networks where momentum-based methods HB and NAG share similar implicit bias with GD, they exhibit different implicit bias for diagonal linear networks compared to GD in two main aspects:

- Compared to GF, HB and NAG flow also converge to solutions that minimize a norm interpolating between $\ell_2$-norm and $\ell_1$-norm up to initialization scales of parameters, more importantly, HB and NAG flow induce an *extra effect that is equivalent to mitigating the influence of initialization of the model parameters*, which is beneficial for generalization properties of sparse regression (Theorem 1). It is worth to mention that SGD could also yield an initialization mitigation effects (Pesme et al., 2021), although momentum-based methods and SGD modify GD differently. In addition, such benefit of HB and NAG flow depends on hyper-parameters such as initialization and learning rate.

- The solutions of HB and NAG flow also *depend on the initialization of parameters and gradients explicitly and simultaneously*, which may not be benign for the generalization performances for sparse regression, while, in contrast, the solutions of GF only depend on the initialization of parameters.

Therefore, HB and NAG are not always better than GD from the perspective of generalization for sparse regression. Whether HB and NAG have the advantages of generalization over GD is up to the overall effects of the above two distinct effects determined by various hyper-parameters. In particular, when mitigation effects of initialization of parameters brought by HB and NAG outperforms their dependence on the initialization of gradients, HB and NAG will have better generalization performances than GD, e.g., when the initialization is highly biased (Fig. 2(a)), otherwise, they will not show such advantages over GD (Fig. 1(a)). And the difference between GD and HB and NAG highly depends on learning rate and momentum factor, albeit the latter is rather obvious, e.g., the difference disappears for small learning rate.

**Organization.** This paper is organized as follows. In Section 2, we summarize notations, setup, and continuous time approximation modelling details of HB and NAG. Section 3 concentrates on our main results of the implicit bias of momentum-based methods for diagonal linear networks with corresponding numerical experiments to support our theoretical findings. We conclude this work in Section 4. All the proof details and additional experiments are deferred to Appendix.

RELATED WORKS

The study of implicit bias started from Soudry et al. (2018) where GD has been shown to return the max-margin classifier for the logistic-regression problem. The analysis for classification problems was then generalized to linear networks (Ji & Telgarsky, 2019), more general homogeneous networks (Lyu & Li, 2020; Chizat & Bach, 2020), and other training strategies (Lyu & Zhu, 2022) for homogeneous networks. For regression problems, Li et al. (2021) showed that gradient flow for matrix factorization implicitly prefers the low-rank solution. Azulay et al. (2021); Yun et al. (2021) studied the implicit bias of GD for standard linear networks. For the diagonal linear networks, Azulay et al. (2021); Yun et al. (2021); Woodworth et al. (2020) further revealed the transition from kernel regime (or lazy regime) (Chizat et al., 2019) to rich regime by decreasing the initialization scales from $\infty$ to 0. Besides the full-batch version of gradient descent, Pesme et al. (2021) studied the stochastic gradient flow and showed that the stochastic sampling noise implicitly induces an effect equivalent to reducing the initialization scale, which leads its solution to be closer to the sparse solution compared to gradient flow. Pillaud-Vivien et al. (2020) then analyzed gradient flow with label noise and Even et al. (2023), removing the infinitesimal learning rate approximation, studied the implicit bias of discrete GD and SGD with moderate learning rate for diagonal linear networks. Wang et al. (2022); Gunasekar et al. (2018) showed that momentum-based methods converge to the same max-margin solution as GD for single layer model and linear classification problem. Jelassi & Li (2022) further revealed that momentum-based methods have better generalization performance than GD for classification problem. Ghosh et al. (2023) conducted a model-agnostic analysis of $\mathcal{O}(\eta^2)$ continuous approximate version of HB and also showed the generalization advantages of HB.

As stated early, the most famous momentum-based methods perhaps are HB (Polyak, 1964) and NAG (Nesterov, 1983). Besides the early applications of momentum-based methods in the convex optimization literature, Rumelhart et al. (1986) firstly applied HB to the training of deep learning models. The recent work (Sutskever et al., 2013) then summarized these momentum-based methods and illustrated their importance in the area of deep learning. Wibisono et al. (2017) demonstrated that SGD, HB and NAG generalize better than adaptive methods such as Adam (Kingma & Ba, 2017) and AdaGrad (Duchi et al., 2011) for deep networks by conducting experiments on classification problems. To characterize the properties of momentum-based methods, continuous time approximations of them are introduced in several recent works. Su et al. (2014) provided a second-order ODE to precisely describe the NAG with momentum factor depending on the iteration count. Wilson et al. (2016) derived a limiting equation for both HB and NAG when the momentum factor depends on learning rate or iteration count. Shi et al. (2018) further developed high-resolution limiting equations for HB and NAG, and Wibisono et al. (2016) designed a general framework from the perspective of Bregman Lagrangian. When the momentum is fixed and does not depend on learning rate or iteration count, Kovachki & Stuart (2021) developed the continuous time approximation, the modified equation in the numerical analysis literature, for HB and NAG.

Compared to previous works, we develop the continuous time approximations of HB and NAG for *deep learning models and regression problems*, and we focus on the *implicit bias of HB and NAG* rather than GD (Woodworth et al., 2020). Therefore, we emphasize that our results are novel

and require new techniques due to the second-order ODE nature of dynamics of momentum-based methods, which is different from the first-order ODE of gradient flow.

## 2  PRELIMINARIES

**Notations.**  We let $\{1, \ldots, L\}$ be all integers between 1 and $L$. The dataset with $n$ samples is denoted by $\{(x_i, y_i)\}_{i=1}^{n}$, where $x_i \in \mathbb{R}^d$ is the $d$-dimensional input and $y_i \in \mathbb{R}$ is the scalar output. The data matrix is represented by $X \in \mathbb{R}^{n \times d}$ where each row is a feature $x_i$ and $y = (y_1, \ldots, y_n)^T \in \mathbb{R}^n$ is the collection of $y_i$. For a vector $a \in \mathbb{R}^d$, $a_j$ denotes its $j$-th component and its $\ell_p$-norm is $\|a\|_p$. For a vector $a(t)$ depending on time, we use $\dot{a}(t) = da/dt$ to denote the first time derivative and $\ddot{a}(t) = d^2a/dt^2$ for the second time derivative. The element-wise product is denoted by $\odot$ such that $(a \odot b)_j = a_j b_j$. We let $\mathbf{e}_d = (1, \ldots, 1)^T \in \mathbb{R}^d$. For a square matrix $W \in \mathbb{R}^{d \times d}$, we use $\mathrm{diag}(W)$ to denote the corresponding vector $(W_{11}, \ldots, W_{dd})^T \in \mathbb{R}^d$.

**Heavy-Ball and Nesterov's method of accelerated gradients.**  Heavy-Ball (HB) and Nesterov's method of accelerated gradients (NAG) are perhaps the most widely adopted momentum-based methods. Different from GD, HB and NAG apply a two-step scheme (Sutskever et al., 2013). In particular, for Eq. (1) let $k$ be the iteration number, $\mu$ be the momentum factor, $\eta$ be the learning rate, and $p \in \mathbb{R}^d$ be the momentum of parameter $\beta$, then HB updates $\beta$ as follows:

$$p_{k+1} = \mu p_k - \eta \nabla L(\beta_k), \ \beta_{k+1} = \beta_k + p_{k+1} \tag{2}$$

where $p_0 = 0$. Similarly, NAG can also be written as a two-step manner

$$p_{k+1} = \mu p_k - \eta \nabla L(\beta_k + \mu p_k), \ \beta_{k+1} = \beta_k + p_{k+1} \tag{3}$$

with $p_0 = 0$. Note that although previous works (Wilson et al., 2016; Su et al., 2014; Nesterov, 2014; Shi et al., 2018) considered HB and NAG with momentum factor depending on the learning rate $\eta$ or iteration count $k$, HB and NAG are generally implemented with constant momentum factor such as in PyTorch (Paszke et al., 2017). Therefore a constant momentum factor $\mu$ is assumed in this work as in Kovachki & Stuart (2021) to be consistent with such practice. We emphasize that this choice, the fixed momentum factor independent of iteration count or learning rate, is not at odds with previous works. Instead, we are considering a setting aligning with the widely adopted convention of deep learning tools.

**HB and NAG flow: continuous time approximations of HB and NAG.**  In this work, we analyze the implicit bias of HB and NAG through their continuous time approximations summarized as follows, which provide insights to the corresponding discrete algorithms and enable us to take the great advantages of the convenience of theoretical analysis at the same time.

**Proposition 1** (HB and NAG flow: $\mathcal{O}(\eta)$ approximate continuous time approximations of HB and NAG). *For the model $f(x; \beta)$ with empirical loss function $L(\beta)$, let $\mu \in (0, 1)$ be the fixed momentum factor, the $\mathcal{O}(\eta)$ approximate continuous time limiting equations for the discrete HB (Eq. (2)) and NAG (Eq. (3)) are of the form*

$$\alpha \ddot{\beta} + \dot{\beta} + \frac{\nabla L(\beta)}{1 - \mu} = 0, \tag{4}$$

*where $\eta$ is the learning rate and $\alpha = \frac{\eta(1+\mu)}{2(1-\mu)}$ for HB, and $\alpha = \frac{\eta(1-\mu+2\mu^2)}{2(1-\mu)}$ for NAG.*

Eq. (4) follows from Theorem 4 of Kovachki & Stuart (2021) and we present the proof in Appendix B for completeness, where we also discuss the order of approximation. Note that since the learning rate $\eta$ is small, Proposition 1 indicates that, for the model parameter $\beta$, modifying GD with fixed momentum is equivalent to perturb the re-scaled gradient flow equation $d\beta/dt = \nabla L(\beta)/(1-\mu)$ by a small term proportional to $\eta$. More importantly, this modification term offers us considerably more qualitative understanding regarding the dynamics of momentum-based methods, which will become more significant for large learning rate—a preferable choice in practice.

**Over-parameterized regression.**  We consider the regression problem for the $n$-sample dataset $\{(x_i, y_i)\}_{i=1}^{n}$ where $n < d$ and assume the existence of the perfect solution, i.e., there exist interpolation solutions $\beta^*$ such that $x_i^T \beta^* = y_i$ for any $i \in \{1, \ldots, n\}$. For the parametric model

$f(x; \beta) = \beta^T x$, we use the quadratic loss $\ell_i = (f(x_i; \beta) - y_i)^2$ and the empirical loss $L(\beta)$ is

$$L(\beta) = \frac{1}{2n} \sum_{i=1}^{n} \ell_i(\beta) = \frac{1}{2n} \sum_{i=1}^{n} (f(x_i; \beta) - y_i)^2. \tag{5}$$

**Diagonal linear networks.** The diagonal linear network (Woodworth et al., 2020) is a popular proxy model for deep neural networks. It corresponds to an equivalent linear predictor $f(x; \beta) = \theta^T x$, where $\theta = \theta(\beta)$ is parameterized by the model parameters $\beta$. For the diagonal linear networks considered in this paper, we study the 2-layer diagonal linear network, which corresponds to the parameterization[1] of $\theta = u \odot u - v \odot v$ in the sense that

$$f(x; \beta) := f(x; u, v) = (u \odot u - v \odot v)^T x, \tag{6}$$

and the model parameters are $\beta = (u, v)$, where $u, v \in \mathbb{R}^d$. We slightly abuse the notation of $L(\theta) = L(\beta)$. Our goal in this paper is to characterize the implicit bias of HB and NAG by precisely *capturing the property of the limit point of $\theta$ and its dependence on various parameters* such as the learning rate and the initialization of parameters for diagonal linear networks $f(x; \beta)$ trained with HB and NAG.

## 3 IMPLICIT BIAS OF HB AND NAG FLOW FOR DIAGONAL LINEAR NETS

To clearly reveal the difference between the implicit bias of (S)GD and momentum-based methods, we start with existing results under the unbiased initialization assumption, and our main result is summarized in Theorem 1 in Section 3.1. We then discuss the dynamics of $\theta$ for diagonal linear networks trained with HB and NAG flow in Section 3.2, which is necessary for the proof of Theorem 1 and may be of independent interest.

For convenience, given a diagonal linear network Eq. (6), let $\xi = (\xi_1, \dots, \xi^d) \in \mathbb{R}^d$ where $\forall i \in \{1, \dots, d\} : \xi_j = |u_j(0)||v_j(0)|$ measures the scale of the initialization, we first present the definition of the unbiased initialization assumed frequently in previous works (Azulay et al., 2021; Woodworth et al., 2020; Pesme et al., 2021).

**Definition 1** (Unbiased initialization for diagonal linear networks). *The initialization for the diagonal linear network Eq. (6) is unbiased if $u(0) = v(0)$, which implies that $\theta(0) = 0$ and $\xi = u(0) \odot v(0)$.*

**Implicit bias of GF.** Recently, Azulay et al. (2021); Woodworth et al. (2020) showed that, for diagonal linear network with parameterization Eq. (6), if the initialization is unbiased (Definition 1) and $\theta(t) = u(t) \odot u(t) - v(t) \odot v(t)$ converges to the interpolation solution, i.e., $\forall i \in \{1, \dots, n\} : \theta^T(\infty) x_i = y_i$, then under gradient flow (GF) $\theta^{\text{GF}}(\infty)$ implicitly solves the constrained optimization problem: $\theta^{\text{GF}}(\infty) = \arg \min_\theta Q_\xi^{\text{GF}}(\theta)$, *s.t.* $X\theta = y$, where $Q_\xi^{\text{GF}}(\theta) = \sum_{j=1}^{d} \left[ \theta_j \operatorname{arcsinh}(\theta_j/(2\xi_j)) - \sqrt{4\xi_j^2 + \theta_j^2} + 2\xi_j \right] / 4$. The form of $Q_\xi^{\text{GF}}(\theta)$ highlights the transition from kernel regimes to rich regimes of diagonal linear networks under gradient flow up to different scales of the initialization: the initialization $\xi \to \infty$ corresponds to the *kernel regime* or *lazy* regime where $Q_\xi^{\text{GF}}(\theta) \propto \|\theta\|_2^2$ and the parameters only move slowly during training, and $\xi \to 0$ corresponds to the *rich regime* where $Q_\xi^{\text{GF}}(\theta) \to \|\theta\|_1$ and the corresponding solutions enjoy better generalization properties for sparse regression. For completeness, we characterize the implicit bias of GF without requiring the unbiased initialization (Definition 1) in the following proposition.

**Proposition 2** (Implicit bias of GF for diagonal linear net with biased initialization). *For diagonal linear network Eq. (6) with biased initialization ($u(0) \neq v(0)$), if $u(t)$ and $v(t)$ follow the gradient flow dynamics for $t > 0$, i.e., $\dot{u} = -\nabla_u L$ and $\dot{v} = -\nabla_v L$, and if the solution converges to the interpolation solution, then*

$$\theta(\infty) = \arg \min_\theta Q_\xi^{\text{GF}}(\theta) + \theta^T \mathcal{R}^{\text{GF}}, \text{ s.t. } X\theta = y \tag{7}$$

*where $\mathcal{R}^{\text{GF}} = (\mathcal{R}_1^{\text{GF}}, \dots, \mathcal{R}_d^{\text{GF}})^T \in \mathbb{R}^d, \forall j \in \{1, \dots, d\} : \mathcal{R}_j^{\text{GF}} = \operatorname{arcsinh}(\theta_j(0)/2\xi_j)/4.$*

Compared to the unbiased initialization case, besides $Q_\xi^{\text{GF}}$, an additional term $\mathcal{R}^{\text{GF}}$ that depends on $\theta(0)$ is required to capture the implicit bias when the initialization is biased.

---

[1] A standard diagonal linear network is $\theta = u \odot v$, which is shown in Woodworth et al. (2020) to be equivalent to our parameterization here. We further discuss this in Appendix C.

### 3.1 IMPLICIT BIAS OF HB AND NAG

Gunasekar et al. (2018); Wang et al. (2022) argued that momentum does not change the implicit bias of GF for linear regression and classification. For deep neural networks, will modifying GF with the widely adopted momentum change the implicit bias? If it does, will momentum-based methods lead to solutions that have better generalization properties? In the following, we characterize the implicit bias of HB and NAG flow (Proposition 1) for diagonal linear networks to compare with that of GF and answer these questions. For completeness, we do not require the unbiased initialization $u(0) = v(0)$ condition and let $\exp(a) \in \mathbb{R}^d$ denote the vector $(e^{a_1}, \dots, e^{a_d})^T$ for a vector $a \in \mathbb{R}^d$. Recall that $\xi = (\xi_1, \dots, \xi_d)^T \in \mathbb{R}^d$ where $\xi_j = |u_j(0)||v_j(0)|$ measures the scale of the initialization of parameters, we now present our main theorem.

**Theorem 1** (Implicit bias of HB and NAG flow for diagonal linear networks). *For diagonal linear network Eq. (6), let $\mathcal{R} = (\mathcal{R}_1, \dots, \mathcal{R}_d)^T \in \mathbb{R}^d$, then if $u(t)$ and $v(t)$ follow the $\mathcal{O}(\eta)$ approximate continuous version of HB and NAG Eq. (4) for $t \geq 0$ and if the solution $\theta(\infty) = u(\infty) \odot u(\infty) - v(\infty) \odot v(\infty)$ converges to the interpolation solution, then, neglecting all terms of the order $\mathcal{O}(\eta^2)$,*

$$\theta(\infty) = \arg\min_\theta \mathbf{Q}_{\bar\xi(\infty)}(\theta) + \theta^T \mathcal{R}, \ s.t. \ X\theta = y \tag{8}$$

*where*

$$\mathbf{Q}_{\bar\xi(\infty)}(\theta) = \frac{1}{4} \sum_{j=1}^d \left[ \theta_j \operatorname{arcsinh}\left( \frac{\theta_j}{2\bar\xi_j(\infty)} \right) - \sqrt{4\bar\xi_j^2(\infty) + \theta_j^2} + 2\bar\xi_j(\infty) \right],$$

$$\forall j \in \{1, \dots, d\} : \mathcal{R}_j = \frac{1}{4} \operatorname{arcsinh}\left( \frac{\theta_j(0)}{2\xi_j} + \frac{4\alpha\partial_{\theta_j}L(\theta(0))}{1-\mu}\sqrt{1 + \frac{\theta_j^2(0)}{4\xi_j^2}} \right),$$

$$\bar\xi(\infty) = \xi \odot \exp\left(-\alpha\phi(\infty)\right), \ \phi(\infty) = 8(1-\mu)^{-2} \int_0^\infty \nabla_\theta L\left(\theta(s)\right) \odot \nabla_\theta L\left(\theta(s)\right) ds. \tag{9}$$

*Specifically, $\alpha$ is chosen as $\frac{\eta(1+\mu)}{2(1-\mu)}$ if we run HB and $\alpha = \frac{\eta(1-\mu+2\mu^2)}{2(1-\mu)}$ for NAG.*

**Remark.** Theorem 1 is for the HB and NAG flow, which is the order $\mathcal{O}(\eta)$ approximate continuous version of discrete HB and NAG. The $\mathbf{Q}_{\bar\xi(\infty)}$ part for HB and NAG flow has a similar formulation to $Q_\xi^{\text{GF}}$ of GF: both of them are the *hyperbolic entropy* (Ghai et al., 2020). In this sense, the transition from kernel regime to rich regime by decreasing $\xi$ from $\infty$ to 0 also exists for HB and NAG (Appendix C.5). The difference between $\mathbf{Q}_{\bar\xi(\infty)}$ and $Q_\xi^{\text{GF}}$ lies in that HB and NAG flow induce an extra initialization mitigation effect: given $\xi$, $\mathbf{Q}_{\bar\xi(\infty)}$ for HB and NAG flow is equivalent to the hyperbolic entropy of GF with a smaller initialization scale since $\bar\xi(\infty)$ is strictly smaller than $\xi$ due to the fact that $\phi(\infty)$ is a positive integral and finite (Proposition 4). As a result, $\mathbf{Q}_{\bar\xi(\infty)}$ is closer to an $\ell_1$-norm of $\theta$ than $Q_\xi^{\text{GF}}$. Furthermore, compared to the implicit bias of GF when the initialization is biased (Proposition 2), an additional term in $\mathcal{R}$ that depends on the initialization of gradient explicitly is required to capture the implicit bias of HB and NAG flow. The dependence on the initialization of gradient is expected since the first step update of momentum methods simply assigns the initialization of gradient to the momentum, which is crucial for the following updates. Therefore, Theorem 1 takes the first step towards positively answering our fundamental question **(Q)** in the sense that *momentum changes the implicit bias of GD for diagonal linear networks.*

A natural question following the fact that HB and NAG flow induce different implicit bias compared to GF is: will this difference lead to better generalization properties for HB and NAG? The implicit bias of HB and NAG flow is captured by two distinct parts, the hyperbolic entropy $\mathbf{Q}_{\bar\xi(\infty)}$ and $\mathcal{R}$, where the effects of momentum on $\mathbf{Q}_{\bar\xi(\infty)}$ is beneficial for generalization while the effects on $\mathcal{R}$ may hinder the generalization performance and is affected by the biased initialization. Thus the answer highly depends on various conditions. In the following, we present a detailed analysis with corresponding numerical experimental results to answer this question for the case of unbiased initialization and biased initialization, respectively.

### 3.1.1 COMPARISON OF HB/NAG FLOW AND (S)GF FOR UNBIASED INITIALIZATION

When the initialization is unbiased (Definition 1), it is worth to mention that the recent work Pesme et al. (2021) studied the stochastic version of gradient flow, the stochastic gradient flow (SGF), and

revealed that the existence of sampling noise changes the implicit bias of gradient flow in the sense that $\theta^{\text{SGF}}(\infty) = \arg\min_\theta Q^{\text{SGF}}_{\tilde{\xi}_\infty}(\theta)$ under the constraint $X\theta = y$, where

$$Q^{\text{SGF}}_{\tilde{\xi}(\infty)}(\theta) = \sum_{j=1}^d \frac{1}{4}\left[\theta_j \operatorname{arcsinh}\left(\frac{\theta_j}{2\tilde{\xi}_j(\infty)}\right) - \sqrt{4\tilde{\xi}_j^2(\infty) + \theta_j^2} + 2\tilde{\xi}_j(\infty)\right] \qquad (10)$$

with $\tilde{\xi}(\infty)$ being strictly smaller than $\xi$. The remarkable point appears when we compare $\mathbf{Q}_{\bar{\xi}_\infty}$ with $Q^{\text{SGF}}_{\tilde{\xi}(\infty)}$: although SGF and momentum-based methods modify GF differently, i.e., SGF adds stochastic sampling noise while momentum-based methods add momentum to GF, *both of them induce an effect equivalent to reducing the initialization scale!* The difference between them lies in the way how they control such initialization mitigation effect. For SGF this is controlled by the integral of loss function (Pesme et al., 2021), while the effect depends on the integral of gradients for HB and NAG flow since they "accumulate" gradients during training.

To show the difference between (S)GF and momentum-based methods HB and NAG flow, we note that $\mathcal{R}_j$ in Theorem 1 becomes

$$\mathcal{R}_j = \operatorname{arcsinh}\left(\frac{4\alpha(X^Ty)_j}{n(1-\mu)}\right),$$

which is also determined by the dataset, and $\mathcal{R}^{\text{GF}}$ in Proposition 2 is simply zero. Therefore, as long as the initialization of gradients $X^Ty = o(\alpha^{-1}n(1-\mu))$, i.e., $\mathcal{R}$ is small compared to $\mathbf{Q}_{\bar{\xi}(\infty)}$ thus only $\mathbf{Q}_{\bar{\xi}(\infty)}$ matters for characterizing the implicit bias, HB and NAG flow will exhibit better generalization properties for sparse regression due to the initialization mitigation effects of HB and NAG flow that lead $\mathbf{Q}_{\bar{\xi}(\infty)}$ to be closer to the $\ell_1$-norm of $\theta$ than $Q^{\text{GF}}_\xi$. On the other hand, when $\mathcal{R}_j$ is not small compared to $\mathbf{Q}_{\bar{\xi}(\infty)}$, the initialization mitigation effects of HB and NAG flow may not be significant, thus there may not be generalization benefit for HB and NAG flow.

**Numerical Experiments.** To verify these claims, we consider the over-parameterized sparse regression. For the dataset $\{(x_i, y_i)\}_{i=1}^n$ where $x_i \in \mathbb{R}^d$ and $y_i \in \mathbb{R}$, we set $n = 40, d = 100$ and $x_i \sim \mathcal{N}(0, I)$. For $i \in \{1, \ldots, n\}$, $y_i$ is generated by $y_i = x_i^T\theta^*$ where $\theta^* \in \mathbb{R}^d$ is the ground truth solution. We let 5 components of $\theta^*$ be non-zero. Our models are 2-layer diagonal linear networks $f(x; \beta) = u \odot u - v \odot v$. We use $\|\xi\|_1$ to measure the scale of initialization. The initialization of parameters is unbiased by letting $u(0) = v(0) = c\mathbf{e}_d$ where $c$ is a constant and $\|\xi\|_1 = c^2 d$. We consider training algorithms GD, SGD, HB, and NAG. And the generalization performance of the solution for each training algorithm is measured by the distance $D(\theta(\infty), \theta^*) = \|\theta(\infty) - \theta^*\|_2^2$. Since $\mathcal{R}$ is determined by the dataset, to control its magnitude, we build three new datasets $\mathcal{D}_\varepsilon = \{(x_{i;\varepsilon}, y_{i;\varepsilon})\}_{i=1}^d$ where $\forall i \in \{1, \ldots, d\} : x_{i;\varepsilon} = \varepsilon x_i, y_{i;\varepsilon} = \varepsilon y_i$. We then train diagonal linear networks using GD and momentum-based methods HB and NAG on each dataset, respectively, and learning rate $\eta = 3 \times 10^{-2}$ and momentum factor $\mu = 0.9$. As shown in Fig. 1, as we decrease the value of $\varepsilon$ which decreases the magnitude of $\mathcal{R}$, the generalization benefit of HB and NAG becomes more significant since their initialization mitigation effects are getting more important. Note that Fig. 1 also reveals the transition to rich regime by decreasing the initialization scales for different training algorithms.

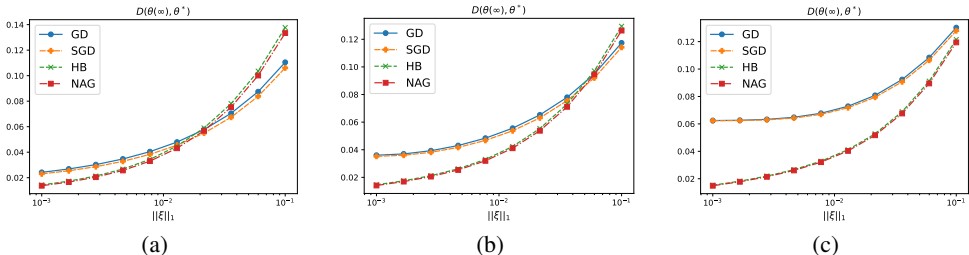

Figure 1: $D(\theta(\infty), \theta^*)$ for diagonal linear networks with unbiased initialization trained with different algorithms and $\varepsilon$ (smaller $\varepsilon$ for smaller $\nabla L(\theta(0))$). **(a).** $\varepsilon = 0.6$. **(b).** $\varepsilon = 0.4$. **(c).** $\varepsilon = 0.2$.

### 3.1.2 COMPARISON OF HB/NAG FLOW AND GF FOR BIASED INITIALIZATION

If the initialization is biased, i.e., $u(0) \neq v(0)$, both the implicit bias of GF and that of HB and NAG flow additionally depends on $\theta(0)$ ($\mathcal{R}^{\text{GF}}$ for GF in Proposition 2 and $\mathcal{R}$ in Theorem 1 for HB

and NAG flow) besides the hyperbolic entropy. Compared to $\mathcal{R}^{\text{GF}}$, $\mathcal{R}$ also includes the explicit dependence on the initialization of gradient that is proportional to $\alpha \nabla L(\theta(0))$. Therefore, recall that $\alpha$ is the order of $\eta$, if $\nabla L(\theta(0)) = o(\alpha^{-1} n(1 - \mu))$ and $\alpha \nabla L(\theta(0))$ is small compared to $\theta(0)$, then $\mathcal{R}^{\text{GF}}$ is close to $\mathcal{R}$, leading to the fact that the difference between the implicit bias of GF and that of HB and NAG flow are mainly due to the initialization mitigation effects of HB and NAG. As a result, we can observe the generalization advantages of HB and NAG over GF (Fig. 2(a)). However, when the initialization is only slightly biased, i.e., $u(0) \neq v(0)$ and $u(0)$ is close to $v(0)$, the dependence on $\nabla L(\theta(0))$ of the solutions of HB and NAG is important and the generalization benefit of HB and NAG for sparse regression may disappear.

**Numerical Experiments.** We use the same dataset $\{(x_i, y_i)\}_{i=1}^{d}$ as in Section 3.1.1. For the hyper-parameters, we set $\eta = 10^{-1}$ and the momentum factor $\mu = 0.9$. To characterize the influence of the extent of the biased part of the initialization, we let $u(0) = \varphi c \mathbf{e}_d$ and $v(0) = \varphi^{-1} c \mathbf{e}_d$ where $\varphi \in (0, 1]$ is a constant measuring the extent of the unbiased part of the initialization. In this way, for any $\varphi$, we have $\xi_j = |u_j(0)||v_j(0)| = c^2$, i.e., the same initialization scale. In order to verify the above theoretical claims, we conduct two sets of experiments: (i). We fix the value of $\varphi$ and train our diagonal linear network with different training algorithms for different scales of initialization $\|\xi\|_1$. As shown in Fig. 2(a), as a result of the initialization mitigation effects, HB, NAG, and SGD exhibit better generalization performance than GD for sparse regression. (ii). We fix $\|\xi\|_1$ and train diagonal linear networks with different biased initialization (different values of $\varphi$). As shown in Fig. 2(b), as we increasing $\varphi$, the initialization becomes less biased and the extra dependence on the initialization of gradient of HB and NAG outperforms their initialization mitigation effects, and, as a result, the generalization benefits of momentum-based methods disappear.

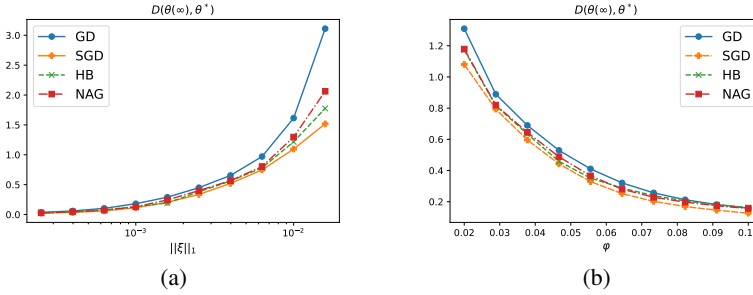

Figure 2: $D(\theta(\infty), \theta^*)$ for diagonal linear networks with biased initialization trained with different algorithms and **(a).** different initialization scales with $\varphi = 0.03$; **(b).** different extents of the biased part of the initialization (smaller $\varphi$ implies that the initialization is more biased) and $\|\xi\|_1 = 0.0046$.

## 3.2   DYNAMICS FOR $\theta$ OF DIAGONAL LINEAR NETWORKS UNDER HB AND NAG FLOW

For diagonal linear networks Eq. (6), dynamics for $\theta$ under HB and NAG flow is crucial to the proof of Theorem 1, and may be of independent interest. Interestingly, different from diagonal linear networks under gradient flow where $\theta$ follows a mirror flow or stochastic gradient flow where $\theta$ follows a stochastic mirror flow with time-varying potential, due to the second-order ODE nature of HB and NAG flow as formulated in Eq. (4), $\theta$ does not directly follow a mirror flow. Instead, HB and NAG flow are special—it is $\theta + \alpha \dot{\theta}$ that follows a mirror flow form with time-varying potential, as shown below.

**Proposition 3** (Dynamics of $\theta$ for diagonal nets trained with HB and NAG flow). *For diagonal linear networks Eq. (6) trained with HB and NAG flow (Eq. (4)) and initialized as $u(0) = v(0)$ and $u(0) \odot u(0) = \xi \in \mathbb{R}^d$, let $\bar{\theta}_\alpha := \theta + \alpha \dot{\theta} \in \mathbb{R}^d$ and its $j$-th component be $\bar{\theta}_{\alpha;j}$, then $\bar{\theta}_\alpha$ follows a mirror flow form with time-varying potential ($\mathcal{R}$ is defined in Theorem 1):*

$$\forall j \in \{1, \ldots, d\}: \; \frac{d}{dt} \nabla \left[ Q_{\xi,j}(\bar{\theta}_\alpha, t) + \bar{\theta}_{\alpha;j} \mathcal{R}_j \right] = -\frac{\partial_{\theta_j} L(\theta)}{1 - \mu}, \tag{11}$$

*where*

$$Q_{\xi,j}(\bar{\theta}_\alpha, t) = \frac{1}{4} \left[ \bar{\theta}_{\alpha;j} \operatorname{arcsinh}\left( \frac{\bar{\theta}_{\alpha;j}}{2 \bar{\xi}_j(t)} \right) - \sqrt{4 \bar{\xi}_j^2(t) + \bar{\theta}_{\alpha;j}} + 2 \bar{\xi}_j(t) \right],$$

$$\bar{\xi}_j(t) = \xi_j e^{-\alpha \phi_j(t)}, \; \phi_j(t) = \frac{8}{(1 - \mu)^2} \int_0^t \partial_{\theta_j} L(\theta(s)) \, \partial_{\theta_j} L(\theta(s)) \, ds, \tag{12}$$

*and $\alpha$ is chosen as in Proposition 1.*

**Remark.** Compared to the mirror flow form of diagonal linear networks under gradient flow $d\nabla Q_\xi^{\mathrm{GF}}(\theta)/dt = -\nabla L(\theta)$, there are three main differences in Eq. (11): (i). It is a second-order ODE since Eq. (11), by noting that $d\bar{\theta}_\alpha/dt = \dot{\theta} + \alpha\ddot{\theta}$, can be written as

$$\alpha\nabla^2 \mathrm{Q}_{\xi,j}(\bar{\theta}_\alpha, t)\ddot{\theta}_j + \nabla^2 \mathrm{Q}_{\xi,j}(\bar{\theta}_\alpha, t)\dot{\theta}_j + \frac{\partial\nabla \mathrm{Q}_{\xi,j}(\bar{\theta}_\alpha, t)}{\partial t} + \frac{\partial_{\theta_j} L(\theta)}{1 - \mu} = 0,$$

while the dynamics of GF is a first-order ODE; (ii). It is $\bar{\theta}_\alpha$, not $\theta$, appears in the mirror flow potential for diagonal linear networks under HB and NAG flow, and an extra term depending on the initialization of gradients is included; (iii). The hyperbolic entropy part of the mirror flow potential $\mathrm{Q}_{\xi,j}(\bar{\theta}_\alpha, t)$ under HB and NAG flow is a time-varying one, and the time-varying part mainly mitigates the influence of the initialization $\xi$ ($\bar{\xi}_j(t) \le \xi$ for any $t \ge 0$).

### 3.3 Effects of Hyper-parameters for Implicit Bias of HB and NAG

As a result of the fact that momentum-based methods (HB and NAG) add a perturbation proportional to the learning rate to time re-scaled gradient flow (as stated in Proposition 1), the difference between their implicit bias depends on learning rate: the limit $\eta \to 0$ leads to $\bar{\xi}(\infty) \to \xi$ and, as a consequence, $\mathbf{Q}_{\bar{\xi}(\infty)} \to Q_\xi^{\mathrm{GF}}$. Therefore, for small learning rate, the implicit bias of momentum-based methods and that of GD are almost the same. This observation coincides with the experience of Rumelhart et al. (1986); Kovachki & Stuart (2021); Ghosh et al. (2023) that setting momentum factor as 0 returns the same solution as reducing the learning rate when momentum factor is non-zero. The discrepancy between the implicit bias of momentum-based methods and that of GD becomes significant for moderate learning rate and momentum factor.

To verify this, we make the initialization biased and $\|\xi\|_1 = 0.1240$, and run: (i). GD with $\eta = 10^{-2}$; (ii). HB and NAG with $\mu = 0.9$ and different $\eta$; (iii). HB and NAG with $\eta = 10^{-2}$ and different $\mu$. We present the generalization performance $D(\theta(t), \theta^*)$ during training for each algorithm with its corresponding training parameters in Fig. 3(a) and Fig. 3(b). Furthermore, we also report the dependence of $D(\theta(\infty), \theta^*)$ on $\eta$ in Fig. 3(c) and the effects of $\mu$ in Fig. 3(d). These results clearly reveal that both decreasing the learning rate and the momentum factor make the difference between the implicit bias of momentum-based methods and that of GD not significant. Experimental details can be found in Appendix A.

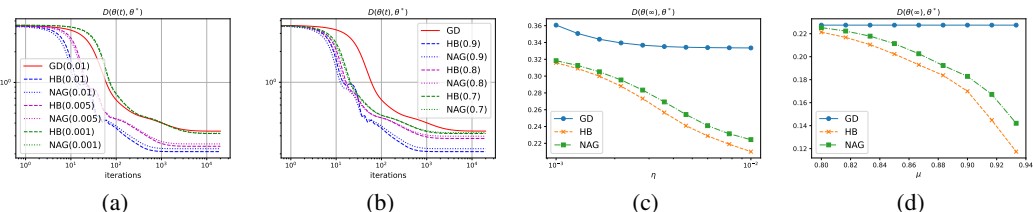

Figure 3: Diagonal nets trained with different algorithms and hyper-parameters: **(a).** $D(\theta(t), \theta^*)$ for different $\eta$ (numbers in the brackets) and $\mu = 0.9$. **(b).** $D(\theta(t), \theta^*)$ for different $\mu$ (numbers in the brackets) and $\eta = 0.01$. **(c).** $D(\theta(\infty), \theta^*)$ for different $\eta$ and $\mu = 0.9$. **(d).** $D(\theta(\infty), \theta^*)$ for different $\mu$ and $\eta = 0.01$.

## 4 Conclusion

In this paper, we have targeted on the unexplored regularization effect of momentum-based methods and we have shown that, *unlike* the single layer linear network, momentum-based methods HB and NAG flow exhibit different implicit bias compared to GD for diagonal linear networks. In particular, we reveal that HB and NAG flow induce an extra initialization mitigation effect similar to SGD that is beneficial for generalization of sparse regression and controlled by the integral of the gradients, learning rate, data matrix, and the momentum factor. In addition, the implicit bias of HB and NAG flow also depends on the initialization of both parameters and gradients explicitly, which may also hinder the generalization, while GD and SGD only depend on the initialization of parameters. It is interesting for future works to explore whether these effects brought by momentum are general across different architectures to further reveal the mysterious properties of momentum-based methods.

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

APPENDIX

In Appendix A, we provide additional numerical experiments to support our theoretical results. In Appendix B, we present the detailed modelling techniques of momentum-based methods HB and NAG. Appendix C presents the proofs for Section 3.

## DETAILED COMPARISON TO RELATED WORKS

**Comparison to Gunasekar et al. (2018); Wang et al. (2022).**   Gunasekar et al. (2018) revealed that there is no difference between the implicit bias of momentum-based methods and that of GD for linear regression problem. In addition, Wang et al. (2022) studied the linear classification problem and showed that momentum-based methods converge to the same max-margin solution as GD for single-layer linear networks, i.e., they share the same implicit bias. These works confirmed that momentum-based methods does not enjoy possible better generalization performance than GD for single-layer models. Compared to these works, our results reveal that momentum-based methods will have different implicit bias when compared to GD for diagonal linear networks, a deep learning models, indicating the importance of the over-parameterization on the implicit bias of momentum-based methods.

**Comparison to Jelassi & Li (2022).**   Jelassi & Li (2022) studied classification problem and also showed that momentum-based methods improve generalization of a linear CNN model partly due to the historical gradients. The setting of our work is different from that of Jelassi & Li (2022): our work focuses on regression problems and diagonal linear networks. In addition, there are also differences between the conclusion of our work and that of Jelassi & Li (2022), in the sense that we conclude that momentum-based methods does not always lead to solutions with better generalization performance than GD, which depends on whether the initialization mitigation effect of momentum-based methods (interestingly this effect can also be regarded as coming from the historical gradients as Jelassi & Li (2022)) outperforms their extra dependence on initialization of gradients. Therefore, the momentum-based method is not always a better choice than GD.

**Comparison to Ghosh et al. (2023).**   The analysis in Ghosh et al. (2023) is general and model-agnostic, in the sense that it did not consider other sources that affect the implicit bias such as model architectures (at least incompletely by only utilizing the calculation of gradients) and initialization, while our work focuses on precisely characterizing the implicit bias of momentum-based methods and its explicit dependence on the architecture and the initialization of both parameters and gradients, which can not be captured solely by the analysis in Ghosh et al. (2023).

## A   ADDITIONAL EXPERIMENTS AND MISSING EXPERIMENTAL DETAILS

### A.1   ADDITIONAL NUMERICAL EXPERIMENTS FOR BIASED INITIALIZATION.

To further characterize the influence of the extent of the biased part of the initialization, we run similar experiments with same hyper-parameters as in Fig. 2(b) except for the scale of the initialization $\|\xi\|_1$. The results are presented in Fig. 4. It can be seen that, for different $\|\xi\|_1$, the generalization benefits of HB and NAG are significant when $\varphi$ is small, i.e., the initialization is highly biased.

For other biased initialization, we consider $u(0) = c\mathbf{e}_d$ for some constant $c \in \mathbb{R}$ and $v(0) = c\mathbf{e}_d + \rho$ for a random gaussian vector $\rho \in \mathbb{R}^d$. We use the same dataset as in Fig. 2(a). As shown in Fig. 5, the generalization performance of HB and NAG solutions become better as we decrease the scale of the initialization, indicating the transition from kernel regime to rich regime. Furthermore, as a result of the initialization mitigation effects, Fig. 5 shows that HB, NAG, and SGD exhibit better generalization performance than GD, which further verifies the benefit of momentum on the generalization when the initialization is biased.

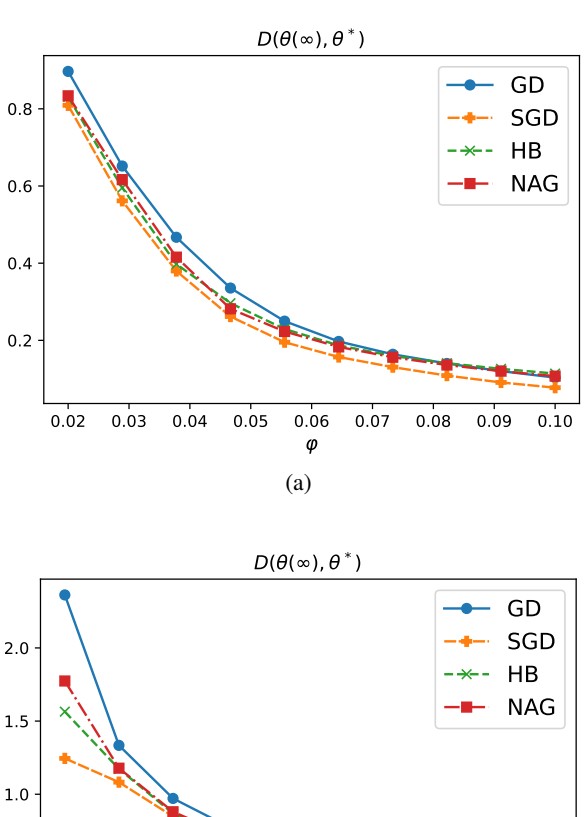

Figure 4: $D(\theta(\infty), \theta^*)$ for diagonal linear networks trained with different algorithms and different values of $\varphi$. **(a).** $\|\xi\|_1 = 0.0022$. **(b).** $\|\xi\|_1 = 0.01$.

## A.2 NON-LINEAR NETWORKS

To explore whether the generalization benefit of momentum-based methods exists for non-linear networks, we conduct experiments for non-linear networks in this section to compare GD with HB and NAG.

**Experiment details for non-linear networks.** We train a four-layer non-linear network $f(x; W)$ with the architecture of $100 \times 100$ `Linear-ReLU`-$100 \times 100$ `Linear-ReLU`-$100 \times 100$ `Linear-ReLU`-$100 \times 1$ `Linear`. The learning rate is fixed as $\eta = 10^{-2}$ and the momentum factor is fixed as $\mu = 0.9$ for HB and NAG. To measure the initialization scales, we vectorize all layer matrices and calculate the sum of $\ell_2$-norm, i.e., we calculate $\sum_{k=1}^{4} \|W_k\|_F^2$ where $W_k$ is the weight matrix of the $k$-th layer. Since non-linear networks are not equivalent to a linear predictor $\theta^T x$ as diagonal linear networks, we sample a newly test data with $\{(x_{i;\text{test}}, y_{i;\text{test}})\}_{i=1}^{40}$ using the ground truth solution $\theta^*$ and the training data distribution and let the test error

$$D = \frac{1}{2n} \sum_{i=1}^{40} \left( f(x_{i;\text{test}}; W) - y_{i;\text{test}} \right)^2$$

measure the generalization performance.

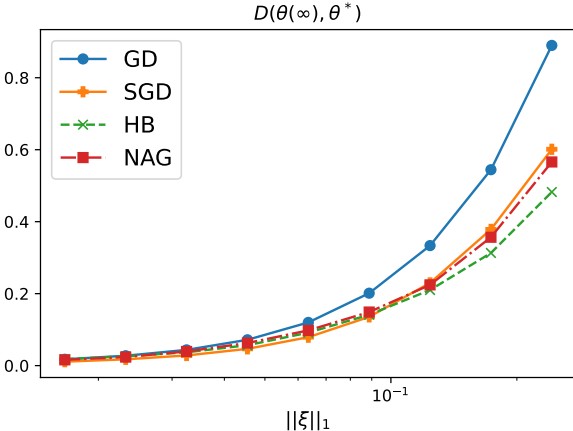

Figure 5: $D(\theta(\infty), \theta^*)$ for diagonal linear networks with biased initialization trained with different algorithms and different values of $\|\xi\|_1$.

We show the benefits of momentum for non-linear networks in the same data of Section 3.1.1 in Fig. 6, which reveals that the benefit of momentum also exists in the non-linear networks, and the test errors are getting lower for smaller initialization scales similar to the diagonal linear networks.

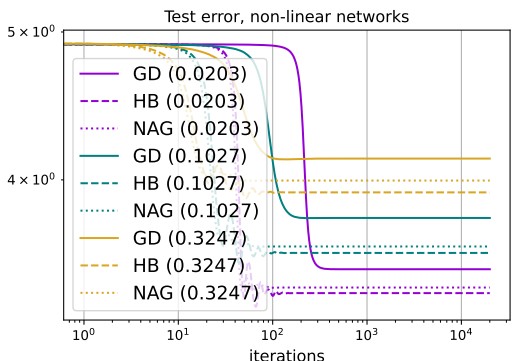

Figure 6: $D(\theta(\infty), \theta^*)$ for non-linear networks trained with different algorithms and different initialization scales (numbers in the bracket)

### A.3 Experimental Details for Fig. 3

The dataset is the same as that in Section 3.1.1. To make the initialization biased, we consider $u(0) = c\mathbf{e}_d$ for some constant $c \in \mathbb{R}$ and $v(0) = c\mathbf{e}_d + \rho$ for a random gaussian vector $\rho \in \mathbb{R}^d$, where we fix $\rho$ for the training of different algorithms with different hyper-parameters.

## B Details of the Continuous Modelling

In Section B.1 we study HB and present the results for NAG in Section B.2.

**Order of convergence of HB and NAG flow.** According to Kovachki & Stuart (2021), Eq. (4) is order $\mathcal{O}(\eta)$ continuous approximate version of the discrete HB and NAG in the sense that, for $k = 0, 1, 2, \ldots$ let $\bar{\beta}_k$ be the sequence given by Eq. (2) or Eq. (3) and let $\beta_k = \beta(k\eta)$ in Proposition 1,

then for any $T \geq 0$, there exists a constant $C = C(T) > 0$ such that

$$\sup_{0 \leq k\eta \leq T} |\beta_k - \bar{\beta}_k| \leq C\eta.$$

### B.1 Continuous time approximation of HB

Recall that the discrete update rule for $\beta$ is

$$p_{k+1} = \mu p_k - \nabla L(\beta_k),$$
$$\beta_{k+1} = \beta_k + \eta p_{k+1},$$

which, noting that $\eta p_k = \beta_k - \beta_{k-1}$, can be further written as a single-step update

$$\beta_{k+1} = \beta_k + \mu(\beta_k - \beta_{k-1}) - \eta \nabla L(\beta). \tag{13}$$

We let time $t = k\eta$ and the continuous version of $\beta_k$ be $\beta(t)$. Note that $\beta(t + \eta) \approx \beta_{k+1}$ and that

$$\beta(t + \eta) = \beta(t) + \eta \dot{\beta}(t) + \frac{\eta^2}{2} \ddot{\beta}(t) + \mathcal{O}(\eta^3),$$

by replacing all $\beta_k$ and $\beta_{k+1}$ with $\beta(t)$ and $\beta(t + \eta)$, respectively, Eq. (13) becomes

$$\beta(t + \eta) - \beta(t) = \mu(\beta(t) - \beta(t - \eta)) - \eta \nabla L(\beta)$$
$$\implies \eta \dot{\beta}(t) + \frac{\eta^2}{2} \ddot{\beta}(t) = \mu \left( \eta \dot{\beta}(t) - \frac{\eta^2}{2} \ddot{\beta}(t) \right) - \eta \nabla L(\beta), \tag{14}$$

which gives us the continuous time approximation of HB:

$$\alpha \ddot{\beta}(t) + \dot{\beta}(t) + \frac{\nabla L(\beta)}{1 - \mu} = 0.$$

with $\alpha = \frac{\eta(1+\mu)}{2(1-\mu)}$.

### B.2 Continuous time approximations of NAG

The discrete update rule for $\beta$ trained with NAG is

$$p_{k+1} = \mu p_k - \eta \nabla L(\beta_k + \mu p_k),$$
$$\beta_{k+1} = \beta_k + p_{k+1},$$

which can also be written as a single-step update

$$\beta_{k+1} = \beta_k + \mu(\beta_k - \beta_{k-1}) - \eta \nabla L(\beta)|_{\beta = \rho_k},$$

where we let

$$\rho_k = \beta_k + \mu p_k.$$

Following similar approach as in the case for HB, this discrete update rule implies that

$$\eta \dot{\beta}(t) + \frac{\eta^2}{2} \ddot{\beta}(t) = \mu \left( \eta \dot{\beta}(t) - \frac{\eta^2}{2} \ddot{\beta}(t) \right) - \eta \nabla L(\beta)|_{\beta = \rho(t)} \tag{15}$$

$$\implies \frac{\eta(1 + \mu)}{2} \ddot{\beta}(t) + (1 - \mu)\dot{\beta}(t) = -\nabla L(\beta)|_{\beta = \rho(t)}. \tag{16}$$

Since the gradient is evaluated at $\beta = \rho(t)$ rather than $\beta(t)$, to further simplify this equation, we note that

$$\rho(t) = \beta(t) + \mu(\beta(t) - \beta(t - \eta))$$
$$= \beta(t) + \eta \mu \dot{\beta}(t) + \mathcal{O}(\eta^3),$$

therefore $\eta \nabla L(\beta)|_{\beta = \rho(t)}$ can be expanded around $\beta(t)$:

$$\eta \nabla L(\beta)|_{\beta = \rho(t)} = \eta \nabla L(\beta)|_{\beta = \beta(t)} + \eta^2 \mu \nabla^2 L(\beta)|_{\beta = \beta(t)} + \mathcal{O}(\eta^2).$$

Meanwhile, by differentiating both sides of Eq. (16) w.r.t $t$, we have

$$\frac{\eta(1+\mu)}{2}\dddot{\beta}(t) + (1-\mu)\ddot{\beta}(t) = -\nabla^2 L(\beta)|_{\beta=\rho(t)}$$

$$\implies \eta^2\mu\nabla^2 L(\beta)|_{\beta=\beta(t)} = -\eta^2(1-\mu)\ddot{\beta}(t)$$

where we multiply $\eta^2$ to both sides of the last equality and omit the terms of the order $\mathcal{O}(\eta^3)$. In this way, Eq. (16) becomes

$$\frac{\eta(1+\mu)}{2}\ddot{\beta}(t) + (1-\mu)\dot{\beta}(t) = -\nabla L(\beta) + \eta\mu(1-\mu)\ddot{\beta}(t)$$

$$\implies \alpha\ddot{\beta} + \dot{\beta} + \frac{\nabla L(\beta)}{1-\mu} = 0$$

with

$$\alpha = \frac{2\mu^2 - \mu + 1}{2(1-\mu)}.$$

### B.3 GENERALIZATION TO $\mathcal{O}(\eta^2)$ APPROXIMATE CONTINUOUS VERSION OF HB

Based on the results of Ghosh et al. (2023), our results presented in this paper could be generalized to the $\mathcal{O}(\eta^2)$ case following exactly the same approach as in the current work, by replacing the current $\mathcal{O}(\eta)$ approximate second-order ODE (Proposition 1) with the $\mathcal{O}(\eta^2)$ approximate version of HB in Ghosh et al. (2023).

## C  PROOFS FOR SECTION 3

In the following, we first discuss the proof sketch for Theorem 1. Then we prove Proposition 3 in Appendix C.1 to show the $\theta$ dynamics and its relation to mirror flow and present the convergence result of the corresponding time-varying potential in Appendix C.2. Finally, we prove Theorem 1 in Appendix C.3. The analysis of the effects of the initialization scale on the implicit bias is presented in Appendix C.5. The proof of Proposition 2 is deferred to Appendix C.6.

**Proof sketch for Theorem 1.**  Since our main result is Theorem 1, we first present the proof sketch. The proof mainly consists of three steps. The first step is to derive the dynamics for $\theta = u \odot u - v \odot v$ of diagonal linear networks for HB and NAG. The second step is to construct the connection of the $\theta$ dynamics with "accelerated" mirror flow (Wilson et al., 2016) with time-varying potential (Proposition 3). Since this time-varying potential converges along training, the third step is simply to apply the optimality condition and derive the implicit bias result.

For diagonal linear networks, the parameterization of $\theta$ is $u \odot u - v \odot v$ and the model parameters are $\beta = (u, v)$. According to Proposition 1, the continuous dynamics of $u$ and $v$ are thus

$$\alpha\ddot{u} + \dot{u} + \frac{\nabla_u L(u,v)}{1-\mu} = 0, \ \alpha\ddot{v} + \dot{v} + \frac{\nabla_v L(u,v)}{1-\mu} = 0, \tag{17}$$

where

$$\alpha = \begin{cases} \frac{\eta(1+\mu)}{2(1-\mu)} & \text{for HB} \\ \frac{\eta(1-\mu+2\mu^2)}{2(1-\mu)} & \text{for NAG} \end{cases}$$

and $\eta$ is the learning rate. Note that since $\alpha$ is the order of $\eta$, we will omit all terms of the order $\mathcal{O}(\eta^2)$. For convenience, we first present several useful properties for the dynamics of HB and NAG for diagonal linear networks Eq. (6) for $\forall j \in \{1, \ldots, d\}$:

1. **Property 1**: $v_j\partial_{u_j}L + u_j\partial_{v_j}L = 0$.

    *Proof.* This is because

    $$\partial_{u_j}L = \frac{2}{n}u_j\sum_{i=1}^n r_ix_{i;j}, \partial_{v_j}L = -\frac{2}{n}v_j\sum_{i=1}^n r_ix_{i;j} \implies v_j\partial_{u_j}L + u_j\partial_{v_j}L = 0 \tag{18}$$

where

$$\frac{1}{n}\sum_{i=1}^{n} r_i x_{i;j} = \partial_{\theta_j} L(\theta) \tag{19}$$

and $r_i = \theta^T x_i - y_i$ is the residual. $\qquad\square$

2. **Property 2:** $\alpha(v_j \dot{u}_j + \dot{v}_j u_j) = \mathcal{O}(\eta^2)$.

   *Proof.* This can be obtained from

   $$\alpha \dot{u}_j = -\alpha\frac{\partial_{u_j} L}{1-\mu} + \mathcal{O}(\eta^2), \ \alpha \dot{v}_j = -\alpha\frac{\partial_{v_j} L}{1-\mu} + \mathcal{O}(\eta^2) \tag{20}$$

   since $\alpha$ is the order of $\eta$, thus, according to Property 1,

   $$\alpha(v_j \dot{u}_j + \dot{v}_j u_j) = -\alpha(v_j \partial_{u_j} L + u_j \partial_{v_j} L) + \mathcal{O}(\eta^2) = \mathcal{O}(\eta^2). \tag{21}$$

   $\qquad\square$

3. **Property 3:** $(u_j + \alpha \dot{u}_j)(v_j + \alpha \dot{v}_j) = u_j v_j + \mathcal{O}(\eta^2)$.

   *Proof.* This can be obtained from

   $$(u_j + \alpha \dot{u}_j)(v_j + \alpha \dot{v}_j) = u_j v_j + \alpha \dot{u}_j v_j + \alpha \dot{v}_j u_j + \alpha^2 \dot{u}_j \dot{v}_j = u_j v_j + \mathcal{O}(\eta^2),$$

   where we use Property 2 in the last equality. $\qquad\square$

4. **Property 4:** $d u_j v_j / dt = 2\alpha \dot{u}_j \dot{v}_j + \mathcal{O}(\eta^2)$.

   *Proof.* To show this, we directly calculate

   $$\begin{aligned}
   \frac{d}{dt} u_j v_j &= \dot{u}_j v_j + \dot{v}_j u_j \\
   &= \left[ -\alpha \ddot{u}_i - \frac{\partial_{u_j} L}{1-\mu} \right] v_j + \left[ -\alpha \ddot{v}_i - \frac{\partial_{v_j} L}{1-\mu} \right] u_j \\
   &= -\alpha(\ddot{u}_i v_j + \dot{u}_j \dot{v}_j + \ddot{v}_i u_j + \dot{u}_j \dot{v}_j) + 2\alpha \dot{u}_j \dot{v}_j \\
   &= -\frac{d}{dt}\left[ \alpha \dot{u}_j v_j + \alpha \dot{v}_j u_j \right] + 2\alpha \dot{u}_j \dot{v}_j \\
   &= 2\alpha \dot{u}_j \dot{v}_j + \mathcal{O}(\eta^2),
   \end{aligned} \tag{22}$$

   where the second line is due to the dynamics of $u$ and $v$ in Eq. (17), and the last equaility is due to Property 2. $\qquad\square$

## C.1 PROOF OF PROPOSITION 3

For convenience, we first recall that, under the conditions of Proposition 3, $\bar{\theta}_{\alpha;j}$ follows a mirror flow form

$$\forall j \in \{1, \ldots, d\} : \ \frac{d}{dt}\left[ \nabla Q_{\xi,j}(\bar{\theta}_\alpha, t) + \theta_j \mathcal{R}_j \right] = -\frac{\partial_{\theta_j} L(\theta)}{1-\mu},$$

where

$$Q_{\xi,j}(\bar{\theta}_\alpha, t) = \frac{1}{4}\left[ \bar{\theta}_{\alpha;j} \operatorname{arcsinh}\left( \frac{\bar{\theta}_{\alpha;j}}{2\bar{\xi}_j(t)} \right) - \sqrt{4\bar{\xi}_j^2(t) + \bar{\theta}_{\alpha;j}} + 2\bar{\xi}_j(t) \right],$$

$$\bar{\xi}_j(t) = \xi_j e^{-\alpha\phi_j(t)}, \ \phi_j(t) = \frac{8}{(1-\mu)^2}\int_0^t \partial_{\theta_j} L(\theta(s)) \partial_{\theta_j} L(\theta(s)) \, ds.$$

Below we prove this result.

*Proof.* The proof consists of two steps: the first step is to derive the dynamics of $\theta$ and the second step is to derive the mirror flow form of the dynamics.

**The dynamics of $\theta$.** We start with the first step. Recall that the parameterization of $\theta_j = u_j^2 - v_j^2$, we conclude that $\theta_j$ follows a second-order ODE different from that of $u$ and $v$ (Eq. (17)) by inspecting the exact expression of $\dot{\theta}_j$

$$\dot{\theta}_j = 2u_j \dot{u}_j - 2v_j \dot{v}_j$$

$$= 2u_j \left( -\alpha \ddot{u}_j - \frac{\partial_{u_j} L}{1 - \mu} \right) - 2v_j \left( -\alpha \ddot{v}_j - \frac{\partial_{v_j} L}{1 - \mu} \right)$$

$$= -2\alpha \left[ u_j \ddot{u}_j + \dot{u}_j \dot{u}_j - v_j \ddot{v}_j - \dot{v}_j \dot{v}_j \right] + 2\alpha \dot{u}_j \dot{u}_j - 2\alpha \dot{v}_j \dot{v}_j - \frac{2 \left( u_j \partial_{u_j} L - v_j \partial_{v_j} L \right)}{1 - \mu}$$

$$= -\alpha \ddot{\theta}_j - \frac{2 \left[ (u_j + \alpha \dot{u}_j) \partial_{u_j} L - (v_j + \alpha \dot{v}_j) \partial_{v_j} L \right]}{1 - \mu}$$

where the second equality is because Eq. (17) and we use Eq. (20) in the last line. Note that if we let

$$G_j = 2 \left[ (u_j + \alpha \dot{u}_j) \partial_{u_j} L - (v_j + \alpha \dot{v}_j) \partial_{v_j} L \right],$$

then the dynamics of $\theta_j$ follows a second-order ODE

$$\alpha \ddot{\theta}_j + \dot{\theta}_j + \frac{G_j}{1 - \mu} = 0, \tag{23}$$

note that although this is similar to the dynamics of $u$ and $v$, they are not the same. To proceed, we need to express $G_j$ with $\theta_j$, which can be done by observing that

$$G_j = 4 \left[ u_j (u_j + \alpha \dot{u}_j) + v_j (v_j + \alpha \dot{v}_j) \right] \partial_{\theta_j} L = 4 H_j \partial_{\theta_j} L$$

where use Eq. (19) in the first equality. Expressing $H_j$ with $\theta_j$ will give us the desired results, which can be done as follows.

$$H_j^2 = \left[ u_j (u_j + \alpha \dot{u}_j) + v_j (v_j + \alpha \dot{v}_j) \right]^2$$

$$= u_j^2 (u_j + \alpha \dot{u}_j)^2 + v_j^2 (v_j + \alpha \dot{v}_j)^2 + 2 v_j u_j (u_j + \alpha \dot{u}_j)(v_j + \alpha \dot{v}_j)$$

$$= u_j^4 + v_j^4 + 2\alpha u_j^3 \dot{u}_j + 2\alpha v_j^3 \dot{v}_j + 2u_j^2 v_j^2 + 2\alpha u_j v_j (u_j \dot{v}_j + v_j \dot{u}_j) + \alpha^2 u_j^2 \dot{u}_j^2$$

$$\quad + \alpha^2 v_j^2 \dot{v}_j^2 + 2\alpha^2 u_j v_j \dot{u}_j \dot{v}_j$$

$$= u_j^4 + v_j^4 + 2u_j^2 v_j^2 + 2\alpha u_j^3 \dot{u}_j + 2\alpha v_j^3 \dot{v}_j + \mathcal{O}(\eta^2) \tag{24}$$

where we use Eq. (21) and $\alpha$ is the order of $\eta$ in the last equality. On the other hand, we observe that the quantity $(\theta_j + \alpha \dot{\theta}_j)^2$ is

$$(\theta_j + \alpha \dot{\theta}_j)^2 = \left[ u_j^2 - v_j^2 + \alpha (2 u_j \dot{u}_j - 2 v_j \dot{v}_j) \right]^2$$

$$= u_j^2 (u_j + 2\alpha \dot{u}_j)^2 + v_j^2 (v_j + 2\alpha \dot{v}_j)^2 - 2 u_j v_j (u_j + 2\alpha \dot{u}_j)(v_j + 2\alpha \dot{v}_j)$$

$$= u_j^4 + v_j^4 + 4\alpha u_j^3 \dot{u}_j + 4\alpha v_j^3 \dot{v}_j - 2u_j^2 v_j^2 + \mathcal{O}(\eta^2) \tag{25}$$

where we use Eq. (21) in the last equality. Combining Eq. (24) and Eq. (25), we have

$$H_j^2 - (\theta_j + \alpha \dot{\theta}_j)^2 = \underbrace{4 u_j^2 v_j^2}_{\clubsuit} - \underbrace{(2\alpha u_j^3 \dot{u}_j + 2\alpha v_j^3 \dot{v}_j)}_{\diamond}, \tag{26}$$

which establishes the relation between $G_j$ and $\theta$. In the following, our goal is to find the relation between $\clubsuit$ and $\diamond$ and $\theta$ to complete the dynamics of $\theta$. Now let $\xi \in \mathbb{R}^d$ and $\xi_j = |u_j(0)||v_j(0)|$ at the initialization[2], then for the term $\clubsuit$, according to Property 4 (Eq. (22)),

$$\frac{d u_j v_j}{dt} = 2\alpha \dot{u}_j \dot{v}_j \tag{27}$$

$$= \frac{2\alpha}{(1 - \mu)^2} \partial_{u_j} L \partial_{v_j} L + \mathcal{O}(\eta^2)$$

$$= -\frac{8\alpha}{(1 - \mu)^2 n^2} u_j v_j \left( \sum_{i=1}^{n} r_i x_{i;j} \right)^2 + \mathcal{O}(\eta^2)$$

$$= -\frac{8\alpha}{(1 - \mu)^2} u_j v_j \partial_{\theta_j} L(\theta) \partial_{\theta_j} L(\theta) + \mathcal{O}(\eta^2) \tag{28}$$

[2]Note that $\xi$ measures the scale of the initialization and $\xi$ becomes $u(0) \odot v(0)$ for unbiased initialization $u(0) = v(0)$. Here we consider the more general biased initialization case.

where we use Eq. (21) in the second equality and Eq. (19) in the third equality. Dividing $u_j v_j$ on both sides and integrating the above equation give us that

$$\ln(u_j v_j) = \ln(u_j(0)v_j(0)) - \frac{8\alpha}{(1-\mu)^2} \int_0^t \partial_{\theta_j} L(\theta(s)) \partial_{\theta_j} L(\theta(s)) ds \tag{29}$$

$$\implies u_j(t)v_j(t) = u_j(0)v_j(0)e^{-\frac{8\alpha}{(1-\mu)^2} \int_0^t \partial_{\theta_j} L(\theta(s)) \partial_{\theta_j} L(\theta(s)) ds}. \tag{30}$$

For ease of notation, we denote

$$\phi_j(t) = \frac{8}{(1-\mu)^2} \int_0^t \partial_{\theta_j} L(\theta(s)) \partial_{\theta_j} L(\theta(s)) ds \geq 0, \tag{31}$$

then ♣ becomes

$$\clubsuit = 4u_j^2(t)v_j^2(t) = 4\xi_j^2 e^{-2\alpha\phi_j(t)} \leq 4\xi_j^2. \tag{32}$$

For the ◇ term, we note that

$$\begin{aligned}
\theta_j \dot\theta_j &= 2(u_j^2 - v_j^2)(u_j \dot u_j - v_j \dot v_j) \\
&= 2\left[ u_j^3 \dot u_j - u_j^2 v_j \dot v_j - v_j^2 u_j \dot u_j + v_j^3 \dot v_j \right] \\
&= 2\left[ u_j^3 \dot u_j + v_j^3 \dot v_j \right] - 2u_j v_j (u_j \dot v_j + v_j \dot u_j),
\end{aligned}$$

which, considering Property 2, further gives us

$$\alpha\theta_j \dot\theta_j = 2\alpha\left[ u_j^3 \dot u_j + v_j^3 \dot v_j \right] + \mathcal{O}(\eta^2).$$

Comparing with the form of ◇, we have

$$\diamondsuit = \alpha\theta_j \dot\theta_j + \mathcal{O}(\eta^2). \tag{33}$$

Combined with the expressions of ♣, $H_j$ can be completely expressed by $\theta$ since Eq. (26) now becomes

$$H_j^2 = (\theta_j + \alpha\dot\theta_j)^2 + 4\xi_j^2 e^{-2\alpha\phi_j(t)} - \alpha\theta_j \dot\theta_j$$

$$\implies H_j = \sqrt{(\theta_j + \alpha\dot\theta_j)^2 + 4\xi_j^2 e^{-2\alpha\phi_j(t)} - \alpha\theta_j \dot\theta_j}. \tag{34}$$

Thus the form of $\theta$ dynamics Eq. (23) is now

$$\frac{1}{4H_j}(\alpha\ddot\theta_j + \dot\theta_j) = -\frac{\partial_{\theta_j} L}{1-\mu}, \tag{35}$$

where $1/H_j$ can be expanded to the order of $\eta$:

$$\begin{aligned}
\frac{1}{H_j} &= \frac{1}{\sqrt{(\theta_j + \alpha\dot\theta_j)^2 + 4\xi_j^2 e^{-2\alpha\phi_j(t)}} \sqrt{1 - \frac{\alpha\theta\dot\theta_j}{(\theta_j + \alpha\dot\theta_j)^2 + 4\xi_j^2 e^{-2\alpha\phi_j(t)}}}} \\
&= \frac{1}{\sqrt{(\theta_j + \alpha\dot\theta_j)^2 + 4\xi_j^2 e^{-2\alpha\phi_j(t)}}} \left( 1 + \frac{1}{2} \frac{\alpha\theta_j \dot\theta_j}{\theta_j^2 + 4\xi_j^2} + \mathcal{O}(\eta^2) \right) \\
&= \frac{1}{\sqrt{(\theta_j + \alpha\dot\theta_j)^2 + 4\xi_j^2 e^{-2\alpha\phi_j(t)}}} + \frac{\alpha}{2} \frac{\theta_j \dot\theta_j}{(\theta_j^2 + 4\xi_j^2)^{\frac{3}{2}}} + \mathcal{O}(\eta^2).
\end{aligned}$$

**Deriving the mirror flow form.** Now we present the second part of the proof. Given $1/H_j$ and its relation with $\theta$, in the following, we are now ready to derive the mirror flow form of $\theta_j$. Note that the L.H.S of Eq. (35) includes a time derivative of $\theta + \alpha\dot\theta$, thus we need to find a mirror flow potential as a function of $\theta + \alpha\dot\theta$, rather than $\theta$. For this purpose, if we define

$$Q_{\xi,j}(\theta + \alpha\dot\theta, t) = q_{\xi,j}(\theta + \alpha\dot\theta, t) + h_j(t)(\theta_j + \alpha\dot\theta_j) \tag{36}$$

such that $q_{\xi,j}$ and $h_j(t)$ satisfy that

$$\nabla^2 q_{\xi,j}(\theta + \alpha\dot\theta, t) = \frac{1}{4\sqrt{(\theta_j + \alpha\dot\theta_j)^2 + 4\xi_j^2 e^{-2\alpha\phi_j(t)}}}, \tag{37}$$

$$\frac{\partial \nabla q_{\xi,j}(\theta + \alpha\dot\theta, t)}{\partial t} + \frac{dh_j(t)}{dt} = \frac{\alpha}{8}\frac{\theta_j\dot\theta_j\dot\theta_j}{(\theta_j^2 + 4\xi_j^2)^{\frac{3}{2}}}, \tag{38}$$

then we will have

$$\frac{d}{dt}\nabla Q_{\xi,j}(\theta + \alpha\dot\theta, t) = \frac{d}{dt}\nabla q_{\xi,j}(\theta + \alpha\dot\theta, t) + \frac{d}{dt}h_j(t)$$

$$= \nabla^2 q_{\xi,j}(\theta + \alpha\dot\theta, t)(\alpha\ddot\theta + \dot\theta) + \frac{\partial \nabla q_{\xi,j}(\theta + \alpha\dot\theta, t)}{\partial t} + \frac{dh_j(t)}{dt},$$

which is exactly the L.H.S of Eq. (35). And we will have the desired mirror flow form of Proposition 3

$$\frac{d}{dt}\nabla Q_{\xi,j}(\theta + \alpha\dot\theta, t) = -\frac{\partial_{\theta_j}L}{1 - \mu}.$$

Therefore, it is now left for us to find $q_{\xi,j}$ and $h_j(t)$ that satisfy Eq. (37) and Eq. (38).

- **Find $q_{\xi,j}(\theta + \alpha\dot\theta, t)$.** Since $q_{\xi,j}(\theta + \alpha\dot\theta, t)$ satisfies Eq. (37), let $\bar\xi_j(t) = \xi_j e^{-\alpha\phi_j(t)}$, we integrate both sides of Eq. (37) to obtain that

$$\nabla q_{\xi,j}(\theta + \alpha\dot\theta, t) = \int \frac{d(\theta_j + \alpha\dot\theta_j)}{4\sqrt{(\theta_j + \alpha\dot\theta_j)^2 + 4\bar\xi_j^2(t)}}$$

$$= \frac{\ln\left(\sqrt{(\theta_j + \alpha\dot\theta_j)^2 + 4\bar\xi_j^2(t)} + (\theta_j + \alpha\dot\theta_j)\right)}{4} + C. \tag{39}$$

To determine the constant $C$, we require that

$$\nabla Q_{\xi,j}(\theta(0) + \alpha\dot\theta(0), 0) = 0, \tag{40}$$

which gives us $\nabla q_{\xi,j}(\theta(0) + \alpha\dot\theta(0), 0) + h_j(0) = 0$. Let $\Delta_j = \theta_j(0) + \alpha\dot\theta_j(0)$ and note that $\bar\xi_j(0) = \xi_j$, we can determine the constant $C$ as

$$C = -\frac{\ln\left(\sqrt{(\theta_j(0) + \alpha\dot\theta_j(0))^2 + 4\bar\xi_j^2(0)} + (\theta_j(0) + \alpha\dot\theta_j(0))\right)}{4} - h_j(0)$$

$$= -\frac{\ln\left[2\xi_j\left(\sqrt{1 + \frac{\Delta_j^2}{4\xi_j^2}} + \frac{\Delta_j}{2\xi_j}\right)\right]}{4} - h_j(0)$$

$$= -\frac{\ln(2\xi_j)}{4} - D_{\xi_j,\Delta_j} - h_j(0) \tag{41}$$

where

$$D_{\xi_j,\Delta_j} = \frac{\ln\left(\sqrt{1 + \frac{\Delta_j^2}{4\xi_j^2}} + \frac{\Delta_j}{2\xi_j}\right)}{4} = \frac{1}{4}\text{arcsinh}\left(\frac{\Delta_j}{2\xi_j}\right).$$

Therefore, $\nabla q_{\xi,j}(\theta + \alpha\dot\theta, t)$ should satisfy that

$$\nabla q_{\xi,j}(\theta + \alpha\dot\theta, t)$$

$$= \frac{\ln\left(\sqrt{(\theta_j + \alpha\dot\theta_j)^2 + 4\bar\xi_j^2(t)} + (\theta_j + \alpha\dot\theta_j)\right) - \ln(2\xi_j)}{4} - D_{\xi_j,\Delta_j} - h_j(0). \tag{42}$$

The form of $q_{\xi,j}$ can be obtained by solving the above equation. For convenience, we replace all $\theta_j + \alpha\dot\theta_j$ with a variable $x$ in the above equation and solve

$$
\begin{aligned}
\nabla q_{\xi,j}(x,t) &= \frac{1}{4}\ln\left(\frac{\sqrt{x^2 + 4\bar\xi_j^2(t)} + x}{2\xi_j}\right) - D_{\xi_j,\Delta_j} - h_j(0) \\
&= \frac{1}{4}\ln\left(\frac{\sqrt{x^2 + 4\bar\xi_j^2(t)} + x}{2\xi_j e^{-\alpha\phi_j(t)}}\right) + \frac{\ln(e^{-\alpha\phi_j(t)})}{4} - D_{\xi_j,\Delta_j} - h_j(0) \\
&= \frac{1}{4}\ln\left(\sqrt{\frac{x^2}{4\bar\xi_j^2(t)} + 1} + \frac{x}{2\bar\xi_j(t)}\right) - \frac{\alpha\phi_j(t)}{4} - D_{\xi_j,\Delta_j} - h_j(0) \\
&= \frac{1}{4}\operatorname{arcsinh}\left(\frac{x}{2\bar\xi_j(t)}\right) - \frac{\alpha\phi_j(t)}{4} - D_{\xi_j,\Delta_j} - h_j(0).
\end{aligned}
\tag{43}
$$

Integrating both sides of the above equation directly gives us that $q_{\xi,j}(x,t)$ has the form of

$$
\begin{aligned}
&q_{\xi,j}(x,t) \\
&= \frac{1}{4}\int \operatorname{arcsinh}\left(\frac{x}{2\bar\xi_j(t)}\right) dx - \frac{\alpha\phi_j(t)x}{4} - D_{\xi_j,\Delta_j}x - h_j(0)x \\
&= \frac{2\bar\xi_j(t)}{4}\left[\frac{x}{2\bar\xi_j(t)}\operatorname{arcsinh}\left(\frac{x}{2\bar\xi_j(t)}\right) - \sqrt{1 + \frac{x}{4\bar\xi_j^2(t)}} + C_1\right] - \frac{\alpha\phi_j(t)x}{4} - D_{\xi_j,\Delta_j}x - h_j(0)x \\
&= \frac{2\bar\xi_j(t)}{4}\left[\frac{x}{2\bar\xi_j(t)}\operatorname{arcsinh}\left(\frac{x}{2\bar\xi_j(t)}\right) - \sqrt{1 + \frac{x^2}{4\bar\xi_j(t)^2}} + 1\right] \\
&\quad - \frac{\alpha x\phi_j(t)}{4} - D_{\xi_j,\Delta_j}x - h_j(0)x,
\end{aligned}
\tag{44}
$$

where we set $C_1 = 1$.

- **Find $h_j(t)$.** The form of $h_j(t)$ can be obtained by solving Eq. (38). According to the form of $\nabla q_j$ in Eq. (42) and the definition of $\phi_j(t)$ in Eq. (31), we need to first calculate $\partial_t \nabla q_{\xi,j}$:

$$
\begin{aligned}
&\partial_t \nabla q_{\xi,j}(\theta + \alpha\dot\theta, t) \\
&= \frac{1}{4}\frac{4\bar\xi_j(t)}{\sqrt{(\theta_j + \alpha\dot\theta_j)^2 + 4\bar\xi_j^2(t)}\left(\sqrt{(\theta_j + \alpha\dot\theta_j)^2 + 4\bar\xi_j^2(t)} + (\theta_j + \alpha\dot\theta_j)\right)}\frac{d\bar\xi_i}{dt} \\
&= -\frac{\alpha\xi_j\bar\xi_j}{\sqrt{(\theta_j + \alpha\dot\theta_j)^2 + 4\bar\xi_j^2(t)}\left(\sqrt{(\theta_j + \alpha\dot\theta_j)^2 + 4\bar\xi_j^2(t)} + (\theta_j + \alpha\dot\theta_j)\right)}\frac{d\phi_j(t)}{dt} \\
&= -\alpha\frac{\xi_j^2}{\sqrt{\theta_j^2 + 4\xi_j^2}\left(\sqrt{\theta_j^2 + 4\xi_j^2} + \theta_j\right)}\frac{8\left(\partial_{\theta_j}L\right)^2}{(1-\mu)^2} + \mathcal{O}(\eta^2).
\end{aligned}
\tag{45}
$$

Putting the above equation back to Eq. (38) immediately gives us that

$$
\begin{aligned}
h_j(t) &= \alpha\int_0^t \frac{\theta_j\dot\theta_j\dot\theta_j}{8(\theta_j^2 + 4\xi_j^2)^{\frac{3}{2}}} + \frac{\xi_j^2}{\sqrt{\theta_j^2 + 4\xi_j^2}\left(\sqrt{\theta_j^2 + 4\xi_j^2} + \theta_j\right)}\frac{8\left(\partial_{\theta_j}L\right)^2}{(1-\mu)^2} ds + C_2 \\
&= \frac{2\alpha}{(1-\mu)^2}\int_0^t \frac{\left(\partial_{\theta_j}L(s)\right)^2}{\sqrt{\theta_j^2(s) + 4\xi_j^2}}\left[\frac{4\xi_j^2}{\sqrt{\theta_j^2(s) + 4\xi_j^2} + \theta_j(s)} + \theta_j(s)\right] ds + C_2 + \mathcal{O}(\eta^2)
\end{aligned}
\tag{46}
$$

where we use

$$
\alpha\dot\theta_j = -4\alpha H_j\frac{\partial_{\theta_j}L}{1-\mu} + \mathcal{O}(\eta^2) = -4\alpha\sqrt{\theta_j^2 + 4\xi_j^2}\frac{\partial_{\theta_j}L}{1-\mu} + \mathcal{O}(\eta^2)
\tag{47}
$$

according to Eq. (35) in the second equality and $C_2 = h_j(0)$ is a constant.

We are now ready to find $Q_{\xi,j}$ by combining the form of $q_{\xi,j}(\theta + \alpha\dot{\theta}, t)$ in Eq. (44) and the form of $h_j(t)$ in Eq. (46), which gives us

$$Q_{\xi,j}(\theta + \alpha\dot{\theta}, t) = \frac{2\bar{\xi}_j(t)}{4}\left[\frac{\theta_j + \alpha\dot{\theta}_j}{2\bar{\xi}_j(t)}\operatorname{arcsinh}\left(\frac{\theta_j + \alpha\dot{\theta}_j}{2\bar{\xi}_j(t)}\right) - \sqrt{1 + \frac{(\theta_j + \alpha\dot{\theta}_j)^2}{4\bar{\xi}_j(t)^2}} + 1\right]$$
$$- \frac{\alpha\theta_j\phi_j(t)}{4} + (\theta_j + \alpha\dot{\theta}_j)h_j(t) - (\theta_j + \alpha\dot{\theta}_j)D_{\xi_j,\Delta_j} - (\theta_j + \alpha\dot{\theta}_j)h_j(0),$$

where, interestingly,

$$-\frac{\alpha\theta_j\phi_j(t)}{4} + (\theta_j + \alpha\dot{\theta}_j)h_j(t) - (\theta_j + \alpha\dot{\theta}_j)h_j(0)$$

$$= -\frac{2\alpha\theta_j}{(1-\mu)^2}\int_0^t (\partial_{\theta_j}L)^2 ds + \frac{2\alpha\theta_j}{(1-\mu)^2}\int_0^t \frac{(\partial_{\theta_j}L(s))^2}{\sqrt{\theta_j^2(s) + 4\xi_j^2}}\left[\frac{4\xi_j^2}{\sqrt{\theta_j^2(s) + 4\xi_j^2} + \theta_j(s)} + \theta_j(s)\right]ds$$

$$= \frac{2\alpha\theta_j}{(1-\mu)^2}\int_0^t \frac{(\partial_{\theta_j}L(s))^2}{\sqrt{\theta_j^2(s) + 4\xi_j^2}}\left[\frac{4\xi_j^2}{\sqrt{\theta_j^2(s) + 4\xi_j^2} + \theta_j(s)} + \theta_j(s) - \sqrt{\theta_j^2(s) + 4\xi_j^2}\right]ds$$

$$= 0. \tag{48}$$

As a result, let $\bar{\theta}_\alpha = \theta + \alpha\dot{\theta}$ and recall that

$$D_{\xi_j,\Delta_j} = \frac{1}{4}\operatorname{arcsinh}\left(\frac{\theta_j(0) + \alpha\dot{\theta}_j(0)}{2\xi_j}\right) \tag{49}$$

where, let $\delta_j = u_j^2(0) - v_j^2(0)$,

$$\theta_j(0) + \alpha\dot{\theta}_j(0) = u_j^2(0) - v_j^2(0) + 2\alpha(u_j(0)\dot{u}_j(0) - v_j(0)\dot{v}_j(0))$$

$$= u_j^2(0) - v_j^2(0) + \frac{4\alpha}{1-\mu}\left[u_j^2(0) + v_j^2(0)\right]\partial_{\theta_j}L(\theta(0))$$

$$= \delta_j + \frac{4\alpha\partial_{\theta_j}L(\theta(0))}{1-\mu}\sqrt{\delta_j^2 + 4\xi_j^2} \tag{50}$$

we have the final form of $Q_{\xi,j}$:

$$Q_{\xi,j}(\bar{\theta}_\alpha, t) = \frac{1}{4}\left[\bar{\theta}_{\alpha;j}\operatorname{arcsinh}\left(\frac{\bar{\theta}_{\alpha;j}}{2\bar{\xi}_j(t)}\right) - \sqrt{4\bar{\xi}_j^2(t) + \bar{\theta}_{\alpha;j}^2} + 2\bar{\xi}_j(t)\right]$$
$$- \frac{1}{4}\bar{\theta}_{\alpha;j}\operatorname{arcsinh}\left(\frac{\delta_j + \frac{4\alpha\partial_{\theta_j}L(\theta(0))}{1-\mu}\sqrt{\delta_j^2 + 4\xi_j^2}}{2\xi_j}\right). \tag{51}$$

The simplest case is when $\delta_j = 0$, i.e., the unbiased initialization with $u(0) = v(0)$ such that $\theta(0) = 0$ and $\nabla_\theta L(\theta(0)) = \frac{1}{n}X^T(X\theta(0) - y) = -\frac{X^Ty}{n}$, then $Q_{\xi,j}(\bar{\theta}_\alpha, t)$ has the form of

$$\frac{1}{4}\left[\bar{\theta}_{\alpha;j}\operatorname{arcsinh}\left(\frac{\bar{\theta}_{\alpha;j}}{2\bar{\xi}_j(t)}\right) - \sqrt{4\bar{\xi}_j^2(t) + \bar{\theta}_{\alpha;j}^2} + 2\bar{\xi}_j(t) + \bar{\theta}_{\alpha;j}\operatorname{arcsinh}\left(\frac{4\alpha(X^Ty)_j}{n(1-\mu)}\right)\right]. \tag{52}$$

Simply redefining

$$Q_{\xi,j}(\bar{\theta}_\alpha, t) = \frac{1}{4}\left[\bar{\theta}_{\alpha;j}\operatorname{arcsinh}\left(\frac{\bar{\theta}_{\alpha;j}}{2\bar{\xi}_j(t)}\right) - \sqrt{4\bar{\xi}_j^2(t) + \bar{\theta}_{\alpha;j}^2} + 2\bar{\xi}_j(t)\right],$$

$$\mathcal{R}_j = \operatorname{arcsinh}\left(\frac{4\alpha(X^Ty)_j}{n(1-\mu)}\right),$$

we finish the proof of Proposition 3:

$$\forall j \in \{1, \ldots, d\}: \quad \frac{d}{dt}\nabla\left[Q_{\xi,j}(\bar{\theta}_\alpha, t) + \bar{\theta}_{\alpha;j}\mathcal{R}_j\right] = -\frac{\partial_{\theta_j}L(\theta)}{1-\mu}.$$

$\square$

## C.2 CONVERGENCE RESULTS FOR $\theta$ DYNAMICS OF HB AND NAG

Since $\mathbf{Q}_{\bar{\xi}_\infty}(\theta)$ in Theorem 1 involves an integral from $t = 0$ to $\infty$, it is necessary to show the convergence of this integral to guarantee the implicit bias result. For this purpose, we establish the convergence result of $\phi(\infty)$ in Theorem 1, whose $j$-th component for $t < \infty$ is $\phi_j(t)$ in Proposition 3. Recall that $Q_{\xi,j}$ is defined in Eq. (12), we have the following proposition.

**Proposition 4** (Convergence of $\phi(\infty)$). *Under the same setting of Theorem 1 and assuming that $\theta(\infty)$ is an interpolation solution, i.e., $X\theta(\infty) = y$, then the integral $\phi(\infty)$ converges and its $j$-th component $\phi_j(\infty)$ satisfies that*

$$\phi_j(\infty) = \frac{16 \left[\mathrm{diag}\left(X^T X\right)\right]_j}{n(1-\mu)} Q_{\xi,j}\left(\theta_j(\infty), 0\right) + C,$$

*where $C = \frac{4\alpha}{n(1-\mu)} \left[\left(\sum_{i=1}^n x_{i;j}^2\right)\left(\sqrt{\theta_j^2(0) + 4\xi_j^2} - 2\xi_j\right) + \sum_{i=1}^n \epsilon_{i;j} \operatorname{arcsinh}\left(\frac{\theta_j(0)}{2\xi_j}\right)\right]$ is a constant, $\epsilon_{i;j} = \left(\sum_{k=1,k\neq j}^d \theta_k(0)x_{i;k} - y_i\right)x_{i;j}$, and $C = 0$ for unbiased initialization $\theta(0) = 0$.*

Typically, solving the integral needs the entire training trajectory of $\theta$, which is hard and equivalent to being aware of the limiting point of $\theta$. From this aspect, Proposition 4 is interesting due to the fact that $\phi(\infty)$ has a rather simple explicit form depending on the data matrix $X^T X$ and $Q(\theta(\infty), 0)$. Furthermore, since the value of $\phi_j(\infty)$ controls the initialization mitigation effects of HB and NAG according to Theorem 1, as an immediate consequence of Proposition 4, such effects depend on learning rate $\eta$ (through the dependence on $\alpha$), the data matrix $X^T X$, initialization $\xi$ (through the dependence on $Q_{\xi,j}$), and the momentum factor $\mu$.

*Proof.* Recall that $\phi_j(t)$ is defined as

$$\phi_j(t) = \frac{8}{(1-\mu)^2} \int_0^t \partial_{\theta_j} L(\theta(s)) \partial_{\theta_j} L(\theta(s)) ds$$

and, according to the dynamics of $\theta$ Eq. (35),

$$\alpha\dot{\theta}_j = -4\alpha\sqrt{\theta_j^2 + 4\xi_j^2} \frac{\partial_{\theta_j} L}{1-\mu} + \mathcal{O}(\eta^2),$$

then we get that $\phi_j(t)$ satisfies

$$\alpha\phi_j(t) = -\frac{4\alpha}{1-\mu} \int_0^t \frac{\partial_{\theta_j} L(\theta(s))}{\sqrt{\theta_j^2(s) + 4\xi_j^2}} \frac{d\theta_j(s)}{ds} ds$$

$$= -\frac{4\alpha}{n(1-\mu)} \int_0^t \frac{\sum_{i=1}^n (\sum_{k=1}^d \theta_k(s)x_{i;k} - y_i)x_{i;j}}{\sqrt{\theta_j^2(s) + 4\xi_i^2}} d\theta_j(s)$$

$$= -\frac{4\alpha(\sum_{i=1}^n x_{i;j}^2)}{n(1-\mu)} \underbrace{\int_0^t \frac{\theta_j(s)}{\sqrt{\theta_j^2(s) + 4\xi_j^2}} d\theta_j(s)}_{\heartsuit}$$

$$- \frac{4\alpha}{n(1-\mu)} \sum_{i=1}^n \underbrace{\int_0^t \frac{\left(\sum_{k=1,k\neq j}^d \theta_k(s)x_{i;k} - y_i\right)x_{i;j}}{\sqrt{\theta_j^2(s) + 4\xi_j^2}} d\theta_j(s)}_{\clubsuit}, \tag{53}$$

where we replace $\alpha\partial_{\theta_j} L$ with $\alpha\dot{\theta}_j$ in the first equality. For the two integral terms, we note that

$$\heartsuit = \frac{1}{2} \int_0^t \frac{1}{\sqrt{\theta_j^2(s) + 4\xi_j^2}} d(\theta_j^2(s) + 4\xi_j^2)$$

$$= \sqrt{\theta_j^2(t) + 4\xi_j^2} + C_1 \tag{54}$$

and, let $\epsilon_{i;j}(t) = \left( \sum_{k=1, k \neq j}^{d} \theta_k(t) x_{i;k} - y_i \right) x_{i;j}$,

$$\clubsuit = \epsilon_{i;j}(t) \operatorname{arcsinh}\left( \frac{\theta_j(t)}{2\xi_j} \right) + C_2. \tag{55}$$

As a result, we obtain the form of $\phi_j(t)$:

$$\phi_j(t) = -\frac{4\alpha}{n(1-\mu)} \left[ (\sum_{i=1}^{n} x_{i;j}^2) \left( \sqrt{\theta_j^2(t) + 4\xi_j^2} - 2\xi_j \right) + \sum_{i=1}^{n} \epsilon_{i;j}(t) \operatorname{arcsinh}\left( \frac{\theta_j(t)}{2\xi_j} \right) \right] + C',$$

where $C'$ is a constant to make $\phi_j(0) = 0$:

$$C' = \frac{4\alpha}{n(1-\mu)} \left[ (\sum_{i=1}^{n} x_{i;j}^2) \left( \sqrt{\theta_j^2(0) + 4\xi_j^2} - 2\xi_j \right) + \sum_{i=1}^{n} \epsilon_{i;j}(0) \operatorname{arcsinh}\left( \frac{\theta_j(0)}{2\xi_j} \right) \right]. \tag{56}$$

Note that when the initialization is unbiased, then we simply have $C' = 0$. Since we assume $\theta$ converges to the interpolation solution, i.e.,

$$\theta^T(\infty) x_i = y_i, \ \forall i \in \{1, \ldots n\},$$

which implies that

$$\sum_{k=1}^{d} \theta_k(\infty) x_{i;k} - y_i = 0, \ \forall i \in \{1, \ldots n\} \implies -\theta_j(\infty) x_{i;j} = \sum_{k=1, k \neq j}^{d} \theta_k(t) x_{i;k} - y_i$$

$$\implies \epsilon_{i;j}(\infty) = -\theta_j(\infty) x_{i;j}^2, \tag{57}$$

we obtain the form of $\phi_j(\infty)$:

$$\alpha \phi_j(\infty) = \frac{4\alpha (\sum_{i=1}^{n} x_{i;j})^2}{n(1-\mu)} \left[ \theta_j(\infty) \operatorname{arcsinh}\left( \frac{\theta_j(\infty)}{2\xi_j} \right) - \sqrt{\theta_j^2(\infty) + 4\xi_j^2} + 2\xi_j \right] + C'$$

$$= \frac{16\alpha (\sum_{i=1}^{n} x_{i;j})^2}{n(1-\mu)} Q_{\xi,j}(\theta(\infty), 0) + C'. \tag{58}$$

$\square$

### C.3 PROOF OF THEOREM 1

In this section, we prove Theorem 1.

*Proof.* If we define

$$\mathbf{Q}_{\bar{\xi}(t)}(\bar{\theta}_\alpha, t) = \sum_{j=1}^{d} Q_{\xi,j}(\bar{\theta}_\alpha, t), \tag{59}$$

then its gradient w.r.t $\bar{\theta}_\alpha$ is

$$\nabla \mathbf{Q}_{\bar{\xi}(t)}(\bar{\theta}_\alpha, t) = \left( \nabla Q_{\xi,1}(\bar{\theta}_\alpha, t), \ldots, \nabla Q_{\xi,d}(\bar{\theta}_\alpha, t) \right)^T,$$

which implies that

$$\frac{d}{dt} \nabla \mathbf{Q}_{\bar{\xi}(t)}(\bar{\theta}_\alpha, t) = \begin{pmatrix} \frac{d}{dt} \nabla Q_{\xi,1}(\bar{\theta}_\alpha, t) \\ \vdots \\ \frac{d}{dt} \nabla Q_{\xi,d}(\bar{\theta}_\alpha, t) \end{pmatrix}$$

$$= -\frac{\nabla L(\theta)}{1-\mu} \tag{60}$$

where we apply Proposition 3 in the second equality. Integrating both sides of the above equation from 0 to $\infty$ gives us

$$\nabla \mathbf{Q}_{\bar{\xi}(\infty)}(\bar{\theta}_\alpha(\infty), \infty) - \nabla \mathbf{Q}_{\bar{\xi}(0)}(\bar{\theta}_\alpha(0), 0) = -\sum_{i=1}^{n} \frac{x_i}{n(1-\mu)} \int_0^{\infty} r_i(s) ds = \sum_{i=1}^{n} x_i \lambda_i. \tag{61}$$

On the other hand, as $t \to \infty$, since we assume $\theta(\infty)$ converges to the interpolation solution, we have, according to Eq. (35),

$$\alpha \dot{\theta}(\infty) \propto \alpha \nabla L(\theta(\infty)) = 0 \implies \bar{\theta}_\alpha(\infty) = \theta(\infty).$$

Considering that $\nabla \mathbf{Q}_{\bar{\xi}(0)}(\bar{\theta}_\alpha(0), 0) = 0$ when we derive the form of $\mathbf{Q}_{\xi,j}$ (Eq. (40)), Eq. (61) implies that

$$\nabla \mathbf{Q}_{\bar{\xi}(\infty)}(\theta(\infty), \infty) = \sum_{i=1}^{n} x_i \epsilon_i. \tag{62}$$

Note that the KKT condition for the optimization problem in Theorem 1 is

$$\nabla \mathbf{Q}_{\bar{\xi}(\infty)}(\theta(\infty), \infty) - \sum_{i=1}^{n} x_i \lambda_i = 0, \tag{63}$$

which is exactly Eq. (62). Thus we finish the proof. $\qquad \square$

## C.4 EQUIVALENCE BETWEEN EQ. (6) AND STANDARD DIAGONAL LINEAR NETWORKS

A standard diagonal linear network is

$$f(x; u, v) = (u \odot v)^T x.$$

If $u$ and $v$ of this model follows the HB and NAG flow (Proposition 1) under the square loss, we note that there is no difference between the forms of their dynamics, since the model $f(x; u, v)$ is completely symmetrical regarding $u$ and $v$, i.e., changing the places of $u$ and $v$ would induce exactly the same model and make no difference. More specifically, the dynamics of $u \odot u$ is

$$\begin{aligned}
\frac{1}{2}\frac{d}{dt}u \odot u = u \odot \dot{u} &= -u \odot \left( \frac{\nabla_u L}{1-\mu} + \alpha \ddot{u} \right) \\
&= -\frac{2}{1-\mu}u \odot (X^T r) \odot v - \left[ \alpha \frac{d}{dt} u \odot \dot{u} - \alpha \dot{u} \odot \dot{u} \right] \\
&= -\frac{2}{1-\mu}u \odot (X^T r) \odot v + \alpha \left[ \frac{2}{1-\mu}\frac{d}{dt}u \odot (X^T r) \odot v + \frac{4(X^T r) \odot v \odot (X^T r) \odot v}{(1-\mu)^2} \right]
\end{aligned} \tag{64}$$

and the dynamics of $v \odot v$ can be obtained by simply replacing all $u$ with $v$ in Eq. (64). Thus if $u(0) \odot u(0) = v(0) \odot v(0)$, i.e., $|u(0)| = |v(0)|$, then we can immediately conclude that $|u(t)| = |v(t)|$ for any $t \geq 0$. Thus we can equivalently parameterize this model with $f(x; u) = (u \odot u)^T x$ further add the weight $v$ such that $f(x; u, v) = (u \odot u - v \odot v)^T x$ can output negative values.

## C.5 ANALYSIS ON THE EFFECTS OF THE INITIALIZATION

For simplicity, we consider the unbiased initialization, and the case for biased initialization is similar. Since the solutions of HB and NAG are composed of two parts, $\mathbf{Q}_{\bar{\xi}(\infty)} = \sum_{j=1}^{d} \mathbf{Q}_{\xi,j}(\bar{\theta}_\alpha, \infty)$ where (note that $\bar{\theta}_\alpha(\infty) = \theta(\infty)$ according to the proof of Theorem 1)

$$\mathbf{Q}_{\xi,j}(\bar{\theta}_\alpha, t) = \frac{1}{4}\left[ \bar{\theta}_{\alpha;j} \operatorname{arcsinh}\left( \frac{\bar{\theta}_{\alpha;j}}{2\bar{\xi}_j(t)} \right) - \sqrt{4\bar{\xi}_j^2(t) + \bar{\theta}_{\alpha;j}^2} + 2\bar{\xi}_j(t) \right]$$

and $\mathcal{R} = (\mathcal{R}_j, \ldots, \mathcal{R}_d)^T \in \mathbb{R}^d$ where

$$\forall j \in \{1, \ldots, d\} : \mathcal{R}_j = \frac{1}{4}\operatorname{arcsinh}\left( \frac{4\alpha(X^T y)_j}{n(1-\mu)} \right),$$

we need to analyze both parts to show the transition from the rich regime to kernel regime, which is different from the case for GD where one only needs to consider the hyperbolic entropy part.

**Small initialization $\xi \to 0$.** We first discuss $Q_{\xi,j}$. When $\xi \to 0$, we have that

$$-\sqrt{4\bar{\xi}_j^2(t) + \bar{\theta}_{\alpha;j}^2} + 2\bar{\xi}_j(t) \to -|\bar{\theta}_{\alpha;j}|$$

and

$$\frac{\bar{\theta}_{\alpha;j}}{2\bar{\xi}_j(t)} \to \text{sign}(\bar{\theta}_{\alpha;j})\infty \implies \bar{\theta}_{\alpha;j} \text{arcsinh}\left(\frac{\bar{\theta}_{\alpha;j}}{2\bar{\xi}_j(t)}\right) \to \text{sign}(\bar{\theta}_{\alpha;j})\bar{\theta}_{\alpha;j}\infty = |\bar{\theta}_{\alpha;j}|\infty,$$

thus $\mathbf{Q}_{\bar{\xi}(\infty)} \to \sum_{j=1}^{d} |\theta_j(\infty)|\infty = \|\theta(\infty)\|_1 \infty$. On the other hand, the $\theta^T \mathcal{R}$ part is finite thus negligible compared to $\mathbf{Q}_{\bar{\xi}(\infty)}$. As a result, we conclude that $\xi \to 0$ corresponds to the rich regime.

**Large initialization $\xi \to \infty$.** For $Q_{\xi,j}$, we note that as $\xi \to \infty$, similar to the case for GD,

$$-\sqrt{4\bar{\xi}_j^2(t) + \bar{\theta}_{\alpha;j}^2} + 2\bar{\xi}_j(t) \to 0$$

and

$$\bar{\theta}_{\alpha;j} \text{arcsinh}\left(\frac{\bar{\theta}_{\alpha;j}}{2\bar{\xi}_j(t)}\right) \to \frac{\bar{\theta}_{\alpha;j}^2}{2\xi_j},$$

thus we obtain that, as $\xi \to \infty$

$$Q_{\xi,j}(\bar{\theta}_\alpha, t) \propto \bar{\theta}_{\alpha;j}^2 \implies \mathbf{Q}_{\bar{\xi}(\infty)} \propto \|\theta(\infty)\|_2^2. \tag{65}$$

On the other hand, $\theta^T \mathcal{R}$ is simply a inner produce between $\theta$ and $\mathcal{R}$. Thus $\mathbf{Q} + \theta^T \mathcal{R}$ is captured by a kernel and $\xi \to \infty$ corresponds to the kernel regime.

## C.6 PROOF OF PROPOSITION 2

*Proof.* Since momentum-based methods is a second-order ODE by adding a perturbation proportional to $\eta$ to the re-scaled GF ODE

$$\dot{u} = -\frac{\nabla_u L}{1-\mu}, \quad \dot{v} = -\frac{\nabla_v L}{1-\mu},$$

Proposition 2 can be proved by following similar steps as the proof of Theorem 1 in Appendix C.3. In particular, let all terms of the order of $\eta$ be zero (thus we ignore the perturbation brought by momentum) and let $\mu = 0$ (thus we make the re-scaled GF a standard GF) in the proof of Theorem 1, we can directly conclude that

$$\theta(\infty) = \arg\min_{\theta} Q(\theta), \quad s.t. \ X\theta = y \tag{66}$$

where

$$Q(\theta) = Q_\xi^{\text{GF}}(\theta) + \theta^T \mathcal{R}^{\text{GF}},$$

$$\mathcal{R}^{\text{GF}} = (\mathcal{R}_1^{\text{GF}}, \ldots, \mathcal{R}_d^{\text{GF}})^T \in \mathbb{R}^d, \ \forall j \in \{1, \ldots, d\} : \mathcal{R}_j^{\text{GF}} = \frac{1}{4} \text{arcsinh}\left(\frac{\theta_j(0)}{2\xi_j}\right). \tag{67}$$

$\square$

