# OpenReview forum: "On the Role of Momentum in the Implicit Bias of Gradient Descent for Diagonal Linear Networks"
_ICLR.cc/2024/Conference — ICLR 2024 Conference Withdrawn Submission_

### Official Review · Reviewer_YbMr · 2023-10-16

**Soundness:** 1 poor
**Presentation:** 3 good
**Contribution:** 1 poor
**Rating:** 5
**Confidence:** 5

**Summary:**

The work explores the role of momentum gradient descent (HB and NAG) in the implicit bias for diagonal linear networks. The authors borrow the $O(\eta)$-close continuous flow approximation for momentum from Kovachki and Stuart and use it to derive the continuous flow for the diaognal linear network. Then they conclude that the modified hyperbolic entropy potentital they find with solving this second order differential equation has a smaller initialization scale. This initialization becomes more small as the accumulation of norm of gradients increase and as the momentum parameter ($\mu$) increase. A standard technique to analyze the implicit bias in such simple networks is the scale of initialization (smaller the richer) which controls the transition from rich to kernel regime (Woodsworth et al). Several works show that SGD and finite step-size also decrease this scale of initialization. And in this paper, the authors try to make an incremental contribution to show that momentum also does the same.

**Strengths:**

The paper is easy to read and follow.

**Weaknesses:**

The paper has serious novelty issues and technical issues. The authors seem to be not aware of the paper [1] which already analyzed the implicit regularization for momentum in deep learning models. In my opinion, the theorem from the paper can be extended to diagonal networks with calcualtions (with almost the same conclusion as theirs). Additionaly the consideration of continuous time approximation for HB and NAG has some technical concerns which I will discuss.

**1) Major novelty issues**

In page-3, the authors claim "Compared to previous works, we develop the continuous time approximations of HB and NAG
for deep learning models, **which is still missing in current literature**, and we focus on the implicit
bias of HB and NAG rather than GD". This statement is False and the paper [1] already analyzed the implicit regularization of momentum through continuous (and modified)  time approximations of HB and NAG for deep learning models. It's unfortunate that the authors were not aware of the work. They [1] do it in a more general setting (not restricted to diagonal networks). This general setting can be also extened to diagonal linear networks with the conclusion unchanged.

**2) A simple extension to the results of [1] may lead to the same conlcusion as this paper**

Consider the theorem 4.1 in [1] which proposes an  $O(\eta^2)$ continuous approximation (as opposed to a weaker O(\eta) approximation in this paper ) to the discrete H.B updates. In this case, the implicit regularization is of the form of the norm of the gradients $\frac{(1+\mu)\eta}{4(1-\mu)^3}|| L(w)||^2$, which promotes flatter sub-trajectories than gradient descent modified flow (which is only $\frac{\eta}{4}|| L(w)||^2$ ). For diagonal linear networks, this regularizer can be explicitly calcualted to be $|| \hat{X}^T r(t) \circ w(t)||^2$ where $w=[u,v]$ according to . I am directly borrowing equation-15 from page-20 in Woodsworth et al, which I feel the authors would be familair with. Now this regularization term  $|| \hat{X}^T r(t) \circ w(t)||^2$ may lead to a form of weighted weight-decay on the factors and it is well known that weight decay on the factors u and v promote sparseness (or minimizes the l1 norm of reconstruction of $\beta$). And this regularization increases with momentum and vanishes for very small step-size (these two are the two main conclusions from this paper). In simpler words, **sparser solutions correspond to the flatter solutions for the diagonal linear networks (see Section 5.2 in https://arxiv.org/pdf/2302.07011.pdf) and [1] already showed that momentum drives trajectories to flatter minimas and this effect vanishes with learning rate tending to 0**.

**3) Major technical concern**

The authors claim "Proposition 1 is an application of Theorem 3 of Kovachki & Stuart (2021) and we present an
alternative proof in Appendix B" where **infact they use exactly Theorem-4 (without any modfiication)**  from Kovachki & Stuart (2021) . Additionally while stating such approximating continuous flows it is very important to mention how much it deviates from the true discrete trajectory (which the original theorem mentions). In this case, the deviation of the approximating continuous flow and the discrete trajectory is order $O(\eta)$ [Theorem 4 in Kovachki & Stuart (2021)]. The problem arises because the authors **do not find the implicit bias of HB and NAG but instead for an approximate continuos version which is $O(\eta)$**. This is problematic because mostly in the paper authors refer to this as "implicit bias of HB and NAG" but it is not. Note that for large time T, this two trajectories deviate as the hidden coefficeint for the $O(\eta)$ is exponential on time. Two questions and discussion can form from this:

*A) Existing works on diagonal linear networks already use continuous version of GD, so why not momentum?*

This is because, the existing works are explicitly for gradient flow and not on gradient descent. The analysis for gradient descent varies largely from those of gradient flow. So, in this current work the implicit bias found is for an $O(\eta)$-approximate continuous version of HB or NAG (and not for HB or NAG)

*B) Why not use an $O(\eta^2)$ continuous approximation for HB?*

If the authors consider the $O(\eta)$-continuous flow as a good approximation for HB,NAG (which it is not), then Theorem-7 in Kovachki and Stuart is more suitable candidate for analysis of HB and NAG. This is because this modified flow is  $O(\eta^2)$ to the trajectory of HB and NAG whereas the continous flow considered is only $O(\eta)$ close. The analysis would also be easier due to the use of first order ODE instead of second order ODE. See point 2 above for details on this.

**4) Observations from section 3.3 is already done in [1]**

The implicit bias for learning rate tending to 0 will lead to similar regurlaizations as GF and this approximate momentum flow is already made in [1] with theorems and experiments.

**5) Missing insights in terms of convergence and saddle to saddle jump issue**

 Momentum is well know to accelerate convergence (the re-sclaed gradient demonstrates that obviously), however the authors claim that momentum will have an effect of smaller effective initializationl. In the class of problems considering incremental learning (diagonal linear networks and matrix sensing), it is well known about the tension between saddle escape time and generalization, smaller initialziation improves generalziation but leads to larger saddle escape time. Does the effective smaller intiialization also increase the saddle escape time ? This is contradictory to the assumption that momentum has faster convergence. The current analysis did not provide any insight.









[1] Avrajit Ghosh, He Lyu, Xitong Zhang, and Rongrong Wang. Implicit regularization in heavyball momentum accelerated stochastic gradient descent. In The Eleventh International Conference on Learning Representations, 2023. URL https://openreview.net/forum?id=
ZzdBhtEH9yB.

**Questions:**

Although there are novelty concerns, I would still like to hear from the authors about the response to the following points

1) Justification for the use of the approximate continuous flow. And why not use a more approximate flow.
2) Is there any viable way the contributions of this work differ from [1]?

---

> ### Author Response · Authors · 2023-11-17
> **Our response, Part I/III**
>
> We thank the reviewer for the detailed comments and valuable suggestions. Below we address your concerns in detail.
>
> ### **Weaknesses part**
> -----
>
> **(1). On "Major novelty issues"**.
>
> We were not aware of the nice related work [1] and we thank the reviewer for the suggestion of it. We include the discussion of this work in the revision and we fix the inaccurate sentence "which is still missing in current literature".
>
> After carefully reading [1], we would like to clarify that we disagree with the reviewer on the aspect of novelty issues. The main reason is that the analysis in [1] is model-agnostic and does not consider other important sources that affect the implicit bias such as model architectures (at least incompletely by only utilizing the calculation of gradients) and initialization, while our work focuses on precisely characterizing the implicit bias of momentum-based methods and its explicit dependence on the architecture and the initialization of both parameters and gradients, which can not be captured solely by the analysis in [1] and support our novelty.
>
> Furthermore, indeed [1] studied HB in a general setting, however, as discussed in the IGR paper [2], only relying on such general analysis is not complete for analyzing the implicit bias. For an instance, the solution of the ODE $\dot{\theta} = -\nabla L(\theta)$ (according to [1, 2], both momentum-based methods and GD can have this form) certainly depends on the initial conditions ––– the initialization of $\theta$ and perhaps the initialization of gradients when considering momentum-based methods. We would like to highlight that overlooking these important sources of implicit bias and directly extending [1] to a more specific setting would lead to an incomplete conclusion that may even contradict the results of numerical  experiments (Fig.1), as we will explain in the next point for the case of diagonal linear networks.
>
> -----
>
> **(2). On "extension of [1] to diagonal linear networks"**.
>
> We appreciate the reviewer for providing this interesting extension.
>
> In our opinion, however, the extension of the results in [1] to diagonal linear networks is incomplete for the following reasons:
>
>   - It can not capture the effects of the initialization of both parameters and gradients on the implicit bias of momentum-based methods, which are important since the transition from kernel to rich regime of diagonal linear networks depends on the initialization of parameters and initialization of gradients are crucial for momentum-based methods, as discussed in our response to the point 1.
>   - To the best of our knowledge, the derived form of weight-decay in the review (which is an $\ell_2$ norm) does not provably promote sparseness of model parameters $u$ and $v$ (which should be described by $\ell_1$ norms). In addition, we would also like to mention that both GD and momentum-based methods can return $\ell_2$-norm solution, which is inconsistent with the argument of the reviewer that the implicit regularization effects $||\nabla L (w)||^2$ of momentum-based methods (the argument can also be applied to GD which also has similar regularization effects) promote sparseness.
>   - It only partly considers the model architecture and can not characterize the exact differences between the properties of solutions for momentum-based methods and that of GD.
>
>  As a consequence, the conclusion that *"sparser solutions correspond to ... to flatter minima[s]"* is not always true. In fact, based on our results in Theorem 1, compared to GD, whether momentum-based methods induce solutions that have better generalization performances for sparse regression depends on the two-fold role of momentum in the implicit bias of GD: the initialization mitigation effect and the explicit dependence on the initialization of gradients.
>
> For example, when the initialization is unbiased (Definition 1 in page 5) and the initialization of gradient is not negligible, momentum-based methods may have worse generalization performance than GD, as shown in Fig.1 where $D ( \theta(\infty), \theta^* )  = || \theta(\infty) - \theta^* ||^2_2$ is larger when the diagonal linear network is trained by momentum-based methods than by GD for certain range of initialization scale $\|\xi\|_1$. Another example is Fig.2(b) where advantages of momentum-based methods over GD on the generalization disappear as the initialization becomes less biased due to the fact that the dependence of solutions of momentum-based methods on the initialization of gradients outperforms the initialization mitigation effects in such case.
>
>  These numerical experiments are not consistent with the conclusion of the simple extension of [1], indicating that it is incomplete to characterize the implicit bias of momentum-based methods solely based on analysis of [1]. Therefore, we believe that such inconsistency between numerical experiments and the incomplete extension of [1] actually support the novelty and significance of our work.

---

> ### Author Response · Authors · 2023-11-17
> **Our response, Part II/III**
>
> **(3). On "Major technical concern"**
>
> We did not check the proof techniques of Theorem 4 in Kovachki and Stuart (2021) when proving Proposition 1, and we thank the reviewer for pointing this similarity out. We fix the corresponding statement in the revision (page 4 of the revision).
>
> We thank the reviewer for the suggestion regarding the importance of stating the deviation of continuous counterpart from the true discrete trajectory. In the revision, we clearly state the $\mathcal{O}(\eta)$ approximate order for the continuous version and fix the corresponding description from "implicit bias of HB and NAG" to "implicit bias of the $\mathcal{O}(\eta)$ approximate continuous version of HB and NAG", or more concretely HB and NAG flow.
>
> - **Regarding Question (A)**: We fix the corresponding description following the suggestion of the reviewer in the revision.
>
> - **Regarding Question (B)**:
>     1. $\mathcal{O}(\eta^2)$ continuous approximation.
>
>         The reason of adopting the $\mathcal{O}(\eta)$ approximate continuous version of momentum-based methods is because we aim to analyze the role of momentum in the implicit bias of widely-studied gradient flow, which is also an order $\mathcal{O}(\eta)$ approximate continuous version of GD, for diagonal linear networks, and the analysis should be conducted in the same order of approximation.
>
>          For the $\mathcal{O}(\eta^2)$ approximate continuous version, based on the results of [1], we believe that our results could be generalized to the $\mathcal{O}(\eta^2)$ case following exactly the same approach as in the current work, by replacing the current $\mathcal{O}(\eta)$ approximate second-order ODE with the $\mathcal{O}(\eta^2)$ approximate version of HB in [1]. We thank the reviewer for the suggestion of $\mathcal{O}(\eta^2)$ continuous approximation and we discuss this in the revision and leave such generalization to future works.
>
>    2. The use of second-order ODE.
>
>        As pointed out by Kovachki and Stuart (2021), the second-order ODE, though being the $\mathcal{O}(\eta)$ approximate continuous version of HB and NAG, contains considerably more qualitative information about the dynamics. In addition, the second-order ODE is also a common choice in the literature for studying momentum-based methods, e.g., Wilson et al. (2016). Therefore, we choose the second-order ODE.
> -----
> **(4). On observations from section 3.3.**
>
> As discussed in our paper, this phenomenon has been observed by, for example, Kovachki and Stuart (2021) and Rumelhart et al. (1986). Section 3.3 mentioned the observations to show that our theoretical results and numerical results match previous ones and further reveal how the implicit bias of momentum-based methods depend on the hyper-parameters. We also state clearly in the revision that [1] also made such kind of observations.
>
> -----
> **(5). On insights in terms of convergence and saddle to saddle jump issue.**
>
> On one hand, whether momentum-based methods indeed accelerate convergence for diagonal linear networks (which is a non-convex problem) is still not clear and the re-scaled gradient perspective suggested by the reviewer is not sufficient since one can simply choose an effective learning rate for GD. To the best of our knowledge,  acceleration effects of momentum-based methods need careful and delicate analysis and many works are for convex problems, e.g., Shi et al. (2018) and Wilson et al. (2016).
>
> On the other hand, as shown in our previous response, momentum-based methods do not always return solutions with better generalization performance than GD.
>
> Therefore, we leave such discussion for future works, since also our focus is the precise characterization of the role of momentum in the implicit bias of gradient flow.
>
> ----
> ### **Questions Part**
>
> 1. Please see our response to point Weakness 3 above.
>
> 2. As discussed in our response to 1 and 2 of the Weakness part above, our work precisely characterizes the implicit bias of momentum-based methods and the corresponding dependence on the initialization of parameters and gradients, and further reveals the conclusion that whether momentum-based methods find solutions with better generalization performance than GD depends on the two-fold effect of momentum. These can not be covered only by the analysis in [1] since it is model-agnostic and does not consider other important sources of implicit bias. Thus it is incomplete, or inconsistent with the numerical experiments, to study the role of momentum on the implicit bias of GD solely based on [1], e.g., it is unclear how the initialization of parameters and gradients can affect the implicit bias of momentum-based methods. We believe that, though being a general analysis, only knowing the preference of flatter trajectory of momentum-based methods than GD ([1]) does not solve the implicit bias of momentum-based methods once and for all.

---

> ### Author Response · Authors · 2023-11-17
> **Our response, Part III/III, Reference**
>
> ### **Reference**
>
> [1] Avrajit Ghosh, He Lyu, Xitong Zhang, and Rongrong Wang. Implicit regularization in heavyball momentum accelerated stochastic gradient descent. ICLR, 2023.
>
> [2] Barrett & Dherin. Implicit Gradient Regularization.
>
> [3] Kovachki & Stuart. Continuous Time Analysis of Momentum Methods, 2021.
>
> [4] Wilson et al. A Lyapunov analysis of momentum methods in optimization, 2016.
>
> [5] Shi et al. Understanding the acceleration phe- nomenon via high-resolution differential equations, 2018.
>
> [6] Rumelhart et al. Learning internal representations by error propagation, 1986.

---

> > ### Comment · Reviewer_YbMr · 2023-11-21
> > **Reviwer response and acknowledgement**
> >
> > I thank the authors for their detailed response and effort in modifying the manuscirpt based on all the reviewer comments. In my opinion, changing the phrase to "finding implicit bias for continuous flow" instead of HB makes the contribution more clear. Also, adding the previous works highlights the contribution better. I have read all the comments to the other reviewers and noted the simmilarity of my concern regarding the $O(\eta)$-flow with reviewer oaZ7. Although the current modificaiton to the paper addresses this joint concern, here are some points which I would like the authors to further clarify:
> >
> > 1) Regarding statement *"The reason of adopting the
> >  approximate continuous version of momentum-based methods is because we aim to analyze the role of momentum in the implicit bias of widely-studied gradient flow, which is also an order
> >  approximate continuous version of GD, for diagonal linear networks, and the analysis should be conducted in the same order of approximation."*
> >
> > **Response**: This is indeed true. but gradient flow is $O(\eta)$-close to gradient descent and this flow is *independent of $\eta$*, however the $O(\eta)$ continuous approximation of HB the authors used is dependent on both $\eta$ and $\mu$. Infact the reason we do not use higher order approximation of gradient descent (https://arxiv.org/abs/2302.01952) is because the modified flow is then a function of $\eta$. But for HB, any continuous flow we chose is a function of $\eta$ and $\mu$ (including the $O(\eta)$ version). In this light, is it not justifiable to use higher order approximations for HB (like $O(\eta^2)$) ? Also commenting "the analysis should be conducted in the same order of approximation" is not entirely correct because of the same reason.
> >
> > 2) Regarding statement *"To the best of our knowledge, the derived form of weight-decay in the review (which is an  L2
> >  norm) does not provably promote sparseness of model parameters u and v"*
> >
> > **Response**: I still respectfully disagree with the authors (please let me know if i misunderstood your argument). It is well known that minimizing $||u||^2 + ||v||^2$ with parameterization $s =u \cdot u - v \cdot v$ leads to minimizing $||s||_{1}$. This has been utilized in many papers in implciit regularization  (see Thoerem 4.2 in https://proceedings.mlr.press/v162/nacson22a.html). The form of $|| \nabla L||^2$ derived in my argument is a rescaled version of weight decay and is indeed has the same form. This goes along the same argument that having weight decay for matrix factors in a low rank matrix recovery makes an explicit penalty on the trace norm of the recovery matrix. And yes, it is true that "GD and momentum-based methods can return
> > l2-norm solution" but that is with large initializaiton. If we still use a weight decay, it will help in biasing to sparse solutions (that is the rich regime can be more easily achieved see for example the explicit regulairzaiotn version in page 6 of https://arxiv.org/pdf/2002.09277.pdf ). Given the response, I would like to restate that I am mentioning sparse solution for the vector s instead of the reparameterized vector u and v  (your statement  "the derived form of weight-decay in the review does not provably promote sparseness of model parameters u and v" seems to suggest that you are referring to sparsity in u and v but i actually mean s).
> >
> > However, I agree with the authors that the analysis in [1] are model and initalization agnostic and this work has similar findings with [1] but also takes into account a very specific and interesting setting which previous works do not. I am eager to increase my score based on these changes. I would still like to hear from the authors regarding the point 2.

---

> ### Author Response · Authors · 2023-11-22
> **Our response on point 1 and 2**
>
> We thank the reviewer so much for the further comment and insightful questions. Below we address your concerns.
>
> ----
> ### **On point 1 of the comment**
>
> We appreciate the reviewer for the argument and we would like to clarify that we were aware of that higher order approximation of GD depends on $\eta$.
>
> We would like to first clarify that, in our response, we did not entirely oppose the reviewer that the $\mathcal{O}(\eta^2)$ continuous approximation for HB is a justifiable choice. In our view, if it is compared to the $\mathcal{O}(\eta^2)$ continuous for GD, e.g., the modified flow in the IGR paper [2], we agree with the reviewer that it is a justifiable choice to characterize the role of momentum on the implicit bias for diagonal linear networks, and it is indeed a promising future direction to discuss this choice.
>
> In this sense, we would like to still emphasize that the same order of approximations for both momentum-based methods and GD should be used *to make a "fair" comparison on their implicit bias for diagonal linear networks*. For an instance, the generalization of the current work (following almost exactly the same technique as in the current one) to $\mathcal{O}(\eta^2)$ approximate version of HB using results of [1] should compare with the generalization of previous gradient flow results for diagonal linear networks [3, 4] to $\mathcal{O}(\eta^2)$ approximate version of GD, not the usual $\mathcal{O}(\eta)$-close version gradient flow, using the modified gradient flow presented in the IGR paper [2].  This is similar to the spirit in [1] where the authors of [1] compared $\mathcal{O}(\eta^2)$ continuous approximation of HB with $\mathcal{O}(\eta^2)$ version of GD (the modified flow), rather than the usual $\mathcal{O}(\eta)$-close version gradient flow.
>
> Therefore, we argue that the main reason why we use the $\mathcal{O}(\eta)$ continuous approximation for HB (and NAG) is that, in this paper, we aim to precisely characterize the the role of momentum in the implicit bias of the widely-studied gradient flow, which is the $\mathcal{O}(\eta)$ continuous approximate version of GD, for diagonal linear networks. And one can naturally generalize the current work to the $\mathcal{O}(\eta^2)$ case following similar techniques, which is a promising future direction.
>
> ----
> ### **On point 2 of the comment**
>
> We thank the reviewer for the clarification *"I am mentioning sparse ... $u$ and $v$"* of the argument in the original comment. And we agree with the reviewer that the derived form of the weight-decay might promote sparseness of $\theta$ ($s$ in the comment of the reviewer).
>
> We would like to kindly clarify that the reason why we mentioned "GD and momentum-based ... $\ell_2$-norm solution" is that only using the model-agnostic analysis for diagonal linear networks, e.g., IGR in [2] and IGR-M in [1], might not capture the transition from rich regime (sparse solution) to kernel regime by tuning the initialization of parameters (perhaps also gradients when considering momentum-based methods), given the argument that the re-scaled weight-decay promotes sparseness of $\theta$. This indicates the importance of studying other sources of implicit bias simultaneously and novelty and significance of our results.
>
> ----
> ### **Reference**
>
> [1] Avrajit Ghosh, He Lyu, Xitong Zhang, and Rongrong Wang. Implicit regularization in heavyball momentum accelerated stochastic gradient descent. ICLR, 2023.
>
> [2] Barrett & Dherin. Implicit Gradient Regularization.
>
> [3] Azulay et al. On the implicit bias of initialization shape: beyond infinitesimal mirror descent.
>
> [4] Woodworth et al. Kernel and rich regimes in overparametrized models.

---

> > ### Comment · Reviewer_YbMr · 2023-11-22
> > **Reviwer response and acknowledgement-2**
> >
> > I thank the authors for the clarifications and ackonwledgement of the arguments provided earlier. I think we are on the same page. Here are my **final comments and summary to the AC** after long discussion :
> >
> > 1) As the authors mentioned, I also agree with them even though the *implicit regularization form derived here and in [1] may take the same form, the analysis in DLNs (this paper) takes into account the model, initialization and the transition from the kernel to the rich regime.* Mentioning this in the manuscript (the form of implicit regualrization), will make the contribution clear and highlight the differences from [1].
> >
> > 2) *The convergence issue HB and comparision with GD still remains unclear.* And how the effective initalization plays a role in convergence, is as proposed by the authors, a future work. In my opinion, understanding the effect of convergence in this setting, is as important as understanding the bias.
> >
> > 3) *I am convinced with the use of first order approximation continuous flow (like GF).* Analyzing implicit biases for the continuous versions GF or HB-flow (to understand discrete counterparts GD or HB) always suffer from exponential deviation in time, the trajectories deviate. In this case, the authors make it clear that they analyze HB-flow or NAG-flow (not the discrete version themselves) and addressed it in the manusrcript.  However, the title of the paper "On the Role of Momentum in the Implicit Bias of Gradient Descent for Diagonal Linear Networks" seems to be misleading as it states gradient descent. I would suggest to make it clear that the analysis holds for continuous versions and the results for GDM would hold during early time (small T).
> >
> > I appreciate the effort of the authors and change my score accordingly based on the reponse.

---

### Official Review · Reviewer_wNfT · 2023-10-29

**Soundness:** 3 good
**Presentation:** 3 good
**Contribution:** 3 good
**Rating:** 6
**Confidence:** 3

**Summary:**

In this manuscript, the authors rigorously examine the implicit biases introduced by momentum-based optimization techniques, specifically focusing on the Heavy-Ball (HB) method and Nesterov's Accelerated Gradient (NAG) scheme, within the context of diagonal linear networks. The paper establishes a dual role for momentum: Firstly, it reveals that both HB and NAG contribute to mitigating the effects of initialization, thereby implicitly steering the linear predictor toward a solution more akin to sparse ($L^1$) regression. Secondly, it uncovers that the implicit bias introduced by momentum is dependent on the gradient initialization, a factor that may not necessarily improve generalization. The claims are verified through numerical experiments.

**Strengths:**

1. The paper is well written and well organized.
2. The paper distinguishes itself by shifting the focus from the extensively studied implicit biases of Gradient Flow (GF) and Gradient Descent (GD) to the role of momentum in neural architectures—a novel and impactful contribution to the literature.
3. Although the explicit characterization of momentum's implicit bias, as articulated in Theorem 1, is intricate—entailing a time integral with undetermined dynamics on $\theta(t)$—its mitigating impact on initialization is nonetheless discernible, given that the integral consistently assumes positive values.
4. The empirical experiments effectively corroborate the theoretical findings.

**Weaknesses:**

1. One potential shortcoming lies in the architectural constraints of the study.  Would it be difficult to generalize the results to multilayer networks that are not diagonal?
2. The paper leaves room for clarification regarding the equivalency of Equation (6) to a "standard" diagonal linear network. Although a footnote references Woodworth et al. (2020), it is beneficial to include this in the paper for completeness.

**Questions:**

Please refer to the previous section

---

> ### Author Response · Authors · 2023-11-17
> **Our response**
>
> We thank the reviewer for the valuable comments and the appreciation of our work. Below we answer your questions.
>
> ### **On Weaknesses Part**
> ----
> 1. **On constraints of the architecture.**
>
>     Diagonal linear networks exhibit many important and interesting properties of complex neural networks such as the transition from kernel regime to rich regime, therefore we choose this insightful model to study the momentum-based methods. Typically the analysis for standard deep linear networks is more complicated. And, based on our results, we believe the generalization to standard deep linear networks, though being harder, are also possible and it is also interesting to investigate whether the initialization mitigation effects of momentum-based methods are general across different architectures.
>
> 2. **On equivalence to standard diagonal linear networks.**
>
>    We thank the reviewer for pointing this out. We add the related discussion of the equivalence between the diagonal structure used in our paper and the standard diagonal linear networks (blue fonts on page 25 in the revision).

---

> ### Comment · Reviewer_wNfT · 2023-11-22
>
> I thank the authors for the response. My rating remains.

---

### Official Review · Reviewer_vt7i · 2023-10-30

**Soundness:** 3 good
**Presentation:** 3 good
**Contribution:** 2 fair
**Rating:** 5
**Confidence:** 4

**Summary:**

This paper extends the analysis of (Woodworth et al., 2020) for the implicit bias over the diagonal linear network from GD to GD with momentum (HB and NAG). Specifically, this paper considers a continuous-time approximation of HB and NAG, and shows that the limiting iteration solves a similar but different minimization problem compared to GD. The authors further analyze the problem and show that  HB and NAG ensures better generalization ability under certain conditions. Experiments are conducted to demonstrate the comparison between HB, NAG, and GD, and the effect of hyperparameter for implicit bias.

**Strengths:**

1. The paper is well-written and well-organized

2. The theoretical results are novel and solid.

**Weaknesses:**

1. **Related works**: There are missing related works. Specifically, this paper claims several times "All these implicit bias works do not consider momentum". However, to my knowledge, there are several existing works studying the implicit bias of momentum-based optimizers, including (Gunasekar et al., 2018; Wang et al., 2022; Jelassi et al., 2022). I strongly suggest the authors discuss the correlation and difference between this paper and the mentioned existing works.

2. **Difference between GD and HB/NAG**: The difference of implicit bias between GD and HB/NAG is still not clear to me. For example, if the $\theta^TR$ term is discarded, isn't the implicit bias of HB/NAG always achieved by GD with a  smaller initialization scale? If that is true, I think it is not proper to say HB/NAG has a better implicit bias than GD, since it seems GD contains as same (if not richer) range of implicit bias as HB/NAG .

3. In the sentence under the second formula in page 7, if letting $\mu=0$, $R$ will equal to $R^{GF}$, which further equals to $0$, right? But I failed to show $R=0$ when $\mu=0$?

4.  About for Figure 1, why GD is better when $||\xi||_1$ is large? Does this contradict the theory?


**Related Works:**

Gunasekar et al., Characterizing Implicit Bias in Terms of Optimization Geometry, 2018

Wang et al., Does Momentum Change the Implicit Regularization on Separable Data?, 2022

Jelassi et al., Towards understanding how momentum improves generalization in deep learning, 2022

**Questions:**

Please see the weakness above.

---

> ### Author Response · Authors · 2023-11-17
> **Our response, Part I/II**
>
> We thank the reviewer so much for the valuable comments and interesting questions. Blow we answer your questions.
>
> ### **Weaknesses Part**
> ----
>
> 1. **On missing related works**.
>
>       We greatly appreciate the reviewer for suggesting these related works. Below we discuss these related works.
>
>       - **Comparison to Gunasekar et al. (2018) and Wang et al. (2022).** We would like to clarify that we discussed Gunasekar et al. (2018) in page 2 (paragraph below the question **Q**) where we mentioned that Gunasekar et al. (2018) revealed that there is no difference between the implicit bias of momentum-based methods and that of GD for linear regression problem. In addition, Wang et al. (2022) studied the linear classification problem and showed that momentum-based methods converge to the same max-margin solution as GD for single-layer linear networks, i.e., they share the same implicit bias. These works confirmed that momentum-based methods does not enjoy possible better generalization performance than GD for single-layer models. Compared to these works, our results reveal that momentum-based methods will have different implicit bias when compared to GD for diagonal linear networks, a deep learning models, indicating the importance of the over-parameterization on the implicit bias of momentum-based methods.
>
>     - **Comparison to Jelassi et al. (2022).** Jelassi et al. (2022) studied classification problem and also showed that momentum-based methods improve generalization of a linear CNN model partly due to the historical gradients. The setting of our work is different from that of  Jelassi et al. (2022): our work focuses on regression problems and diagonal linear networks. In addition, there are also differences between the conclusion of our work and that of Jelassi et al. (2022), in the sense that we conclude that momentum-based methods does not always lead to solutions with better generalization performance than GD, which depends on whether the initialization mitigation effect of momentum-based methods (interestingly this effect can also be regarded as coming from the historical gradients as Jelassi et al. (2022)) outperforms their extra dependence on initialization of gradients. Therefore, the momentum-based method is not always a better choice than GD.
>
>     We add the discussion of these related works in the revision (Page 2, 3, and 12 in the revision) as suggested by the reviewer.
>
> ----
> 2. **On difference between GD and HB/NAG.**
>
>     Indeed, GD contains the same range of implicit bias as HB/NAG. We would like to first clarify that HB/NAG have better implicit bias than GD when their initialization mitigation effects outperform their extra dependence on the initialization of gradients for diagonal linear networks.
>
>     And if this condition is satisfied, when stating that HB/NAG have better implicit bias than GD, we mean that HB/NAG converge to solutions with better generalization performance for sparse regression (since the solutions of momentum-based methods are closer to the $\ell_1$-norm solutions due to the effective initialization mitigation effect) than GD *under exactly the same initialization and architecture*, i.e., if all the other conditions are the same, then solutions returned by HB/NAG generalize better than GD for sparse regression. As an example of indicating such difference between HB/NAG and GD, Wang et al. (2022) showed that momentum-based methods converge to the same max-margin solution as GD for linear classification problem (thus they have same implicit bias), while this is not the case for regression problem with diagonal linear networks --- momentum-based methods converge to different solutions compared to GD, leading to the conclusion that they have different implicit bias.
>
> ----
> 3. **On setting $\mu = 0$ in page 7**.
>
>     We would like to clarify that $\mathcal{R}$ in page 7, which is derived under the unbiased initialization (Definition 1 on page 5), is only for momentum-based methods since it is of the order of $\eta$ which does not exist for GF where no term of the order $\eta$ appears, i.e., $\mathcal{R}^{GF} = 0$ under unbiased initialization $\theta(0) = 0$ (please see Eq.(7) in page 5, or Azulay et al. (2021); Woodworth et al. (2020)).
>
>     The question raised in the review when setting $\mu = 0$ is because only doing so is not sufficient and one should set $\mu = 0$ and neglect all terms of the order $\mathcal{O}(\eta)$ simultaneously to back to the GF equation from the second-order momentum ODE, which leads to $\mathcal{R}$ = 0 in page 7. We would also like to kindly mention that only setting $\mu = 0 $ commonly could not lead various second-order momentum ODEs to be the GF equation, e.g., various high-resolution momentum ODEs in Shi et al. (2018).

---

> ### Author Response · Authors · 2023-11-17
> **Our response, Part II/II**
>
> 4. **On Fig.1**
>
>    We would like to clarify that whether HB/NAG induce solutions with better generalization performance than GD for sparse regression depends on the two-fold role of momentum on the implicit bias: the initialization mitigation effect (beneficial for generalization), which is due to the integral in Eq.(9) on page 6, and the extra explicit dependence on the initialization of gradients (possibly harmful for generalization), which is due to the term of the order $\eta$ in $\mathcal{R}$ in Eq.(8) on page 6.
>
>     When the dependence on the initialization of gradients outperforms the initialization mitigation effect, it is possible that solutions of HB/NAG would generalize worse than that of GD, which is the case of Fig.1(a) when $\theta(0) = 0 $ and $\mathcal{R}^{GF} = 0$. On the other hand, as we decrease the scale of the initialization of gradients, solutions of HB/NAG would generalize better than that of GD, as shown in  Fig.1(c). Therefore Fig.1 does not contradict the theory.
>
>     Furthermore, when the initialization is biased, both $\mathcal{R}$ of HB/NAG and $\mathcal{R}^{GF}$ are nonzero. In this case, as we increase the relative significance of the initialization of gradients as shown in Fig. 2(b) on page 8, the advantages of HB/NAG over GD about the generalization performance gradually disappear, which supports the theory.
>
> ----
> ### **Reference**
>
> Gunasekar et al. Characterizing Implicit Bias in Terms of Optimization Geometry.
>
> Azulay et al. On the implicit bias of initialization shape: beyond infinitesimal mirror descent.
>
> Woodworth et al. Kernel and rich regimes in overparametrized models.
>
> Shi et al. Understanding the acceleration phe- nomenon via high-resolution differential equations.
>
> Wang et al. Does Momentum Change the Implicit Regularization on Separable Data?
>
> Jelassi et al. Towards understanding how momentum improves generalization in deep learning.

---

> > ### Comment · Reviewer_vt7i · 2023-11-22
> > **Acknowledgement after rebuttal**
> >
> > I would like to thank the authors for the detailed response. Still, my concern remains regarding "On difference between GD and HB/NAG": if the same regularization can be achieved by using GD with a smaller initialization, what is the harm of using GD with a smaller initialization? Maybe a worse convergence rate or so? If not, why not just using GD which even saves the memory of momentum?

---

> ### Author Response · Authors · 2023-11-22
> **On difference between GD and HB/NAG**
>
> We thank the reviewer a lot for the reply. Below we address your concern.
>
> Indeed using GD with a smaller initialization might achieve similar regularization effect as HB/NAG. The main harm of using GD with a smaller initialization is the saddle point escape issue. In particular, very
> small initialization scales (especially those required for leaving the kernel regime in wide models) lead to the issue that these scales correspond to the initialization close to a saddle point that might be difficult to escape. This has been discussed in, for example, Nacson et al. (2022).
>
> In our case, to achieve good generalization, a smaller initialization is required for GD compared to HB/NAG, however, this is problematic from an optimization perspective since
> $u = v = 0$ is a saddle point and, as a result, it is highly likely that using smaller initialization
> scale leads to a larger saddle point (the
> vicinity of zero) escape time of GD, which may even not converge when the initialization is too small. As a comparison, HB/NAG could start from a relatively larger initialization than GD to alleviate or avoid the saddle point escape problem to achieve good generalization. This indicates the difference between GD and HB/NAG
>
> ----
> **Reference**
>
> Mor Shpigel Nacson et al. Implicit Bias of the Step Size in Linear Diagonal Neural Networks, ICML 2022.

---

### Official Review · Reviewer_oaZ7 · 2023-11-02

**Soundness:** 1 poor
**Presentation:** 2 fair
**Contribution:** 1 poor
**Rating:** 3
**Confidence:** 4

**Summary:**

The paper studies the implicit regularization effect of the momentum methods through their continuous-time approximations in the case of diagonal networks. The paper discusses the role of initialization and the gradient of the loss at initialization on the recovered implicit bias. The theory is supported by experiments.

**Strengths:**

The paper addresses a well-motivated problem. Given the widespread use of momentum in training deep networks, gaining insights into its implicit regularization and convergence, even for simple non-convex models, is essential.

**Weaknesses:**

I believe that Theorem 1 is incorrect or at least it is incorrectly stated.

The paper studies momentum ODE but the problem is in the analysis, i.e., proof of Proposition 3 and Theorem 1, the $\eta^2$ terms are ignored even after the modelling with an ODE. But the result stated does not state that the implicit bias holds only under this approximation. Hence, the statement in its current form is incorrect.  I am happy to engage in further discussion if the authors bring other points of discussion.

I also do not agree with the proof technique. a) either start with a discrete time algorithm and ignore the higher order terms to recover the potential which it implicitly minimizes or b) use the continuous time counterpart and state the implicit bias for this without any further approximation. In the approach followed by the paper it is not clear what is the order of approximation of the result (the theorem does not even state that it is an approximate result). There are two errors: one stemming from discretization and the other from neglecting second-order terms. It is not evident how close this trajectory to the one described in Theorem 1. However, I strongly feel that the Theorem 1 should be rewritten and it is not acceptable in its current form. I

**Questions:**

already discussed in the weakness part.

---

> ### Author Response · Authors · 2023-11-17
> **Our response**
>
> We thank the reviewer for the valuable comments and questions. Below we address your concerns.
>
> ### **Weaknesses Part**
> ----
>
> **(1). On neglecting second-order terms.**
>
> We would like to clarify that neglecting the second-order terms for the ODE is eligible. This is because the momentum ODE is established to the order of $\eta$, i.e., the momentum ODE in Eq.(4) is in fact $\alpha \ddot{\beta} + \dot{\beta} + \nabla L / (1 - \mu) = \mathcal{O}(\eta^2)$. Thus all terms of order higher than $\eta$ (including the order $\eta^2$) appearing in anywhere else should also be dropped. To be consistent with this convention, the $\eta^2$ terms are ignored in our proof. For example, from Eq.(4) we have
> $ \eta \dot{\beta} = - \eta\frac{\nabla L}{1 - \mu} + \mathcal{O}(\eta^2),$  which is eligible (it states that momentum adds a perturbation of the order $\eta$ to re-scaled gradient flow) and can also be derived from the discrete update rules by following exactly the same approach of the proof for Proposition 1 and ignoring $\eta^2$ terms in the discrete relation Eq.(14).
>
> We mentioned this point in the proof and we thank the reviewer for pointing out that we missed this in the statement of theorems. We fix this in the revision by stating clearly in Proposition 1 that we ignore all terms of the order higher than $\eta$.
>
> One can try to generalize the analysis to include terms of the order higher than $\eta$ by using the momentum ODE with higher order terms, e.g., Eq.(13) on page 11 of Kovachki and Stuart (2021) that includes arbitrarily higher order terms, and following our current approach where one should now preserve these higher order terms. Since most momentum-based modellings of momentum-based methods are truncated to the order of $\eta$, we also focus on this choice.
>
> ----
> **(2). On the order of the continuous approximation of HB and NAG and inaccurate statement of Theorem 1.**
>
> The continuous counterpart of HB and NAG used in our paper (Eq.(4)) is $\mathcal{O}(\eta)$ approximate continuous version of discrete HB and NAG, as shown in Kovachki and Stuart (2021). We thank the reviewer for the suggestion of inaccurate statement of Theorem 1, and we fix this in the revision by carefully and clearly stating the current implicit bias is for the $\mathcal{O}(\eta)$ approximate continuous version of HB and NAG, or more concretely HB and NAG flow, rather than directly for the discrete HB and NAG.
>
> We adopt the $\mathcal{O}(\eta)$ approximate continuous version of momentum-based methods since we aim to analyze the role of momentum in the implicit bias of widely-studied gradient flow, which is also an order $\mathcal{O}(\eta)$ approximate continuous version of GD, for diagonal linear networks, and the analysis should be conducted in the same order of approximation. The analysis can be naturally generalized to the order $\mathcal{O}(\eta^2)$ approximate continuous version of HB by replacing the $\mathcal{O}(\eta)$ approximate continuous version of HB (Proposition 1) with the $\mathcal{O}(\eta^2)$ ones in Kovachki and Stuart (2021) (Eq.(20)) and following similar approach as in the current work.
>
> ----
> ### **Reference**
> Kovachki and Stuart. Continuous Time Analysis of Momentum Methods. arXiv: 1906.04285, 2019.

---

> > ### Comment · Reviewer_oaZ7 · 2023-11-21
> > **Response to author comments**
> >
> > > On neglecting second-order terms. We would like to clarify that neglecting the second-order terms for the ODE is eligible.
> >
> > I disagree with the above statement, you are analysing a continuous time ODE and claiming a guarantee about where it converges to. The ODE might be derived from approximating a discrete process up to some error terms, it does not matter. The result you claim about the limit of the ODE will be false.
> >
> > Even neglecting the second order terms is not done in a correct way. For example, take property 2,3,4 in appendix, to get the implicit bias you integrate the quantities across time and there are no guarantees that this $O(\eta^2) $ residue does not grow unbounded or to $O(1)$. So even your claim that it will be $O(\eta) $ is not correct. Yes, you can counter this argument using the claim about discretization error between GF and GD or NAG flow and NAG. However, the works which say the implicit bias of GF at the infinite time limit are at least consistent for GF (even if you cannot give a bound for discretization).  Hence the technical approach of this work if flawed and not correct.

---

### Meta-Review · Area_Chair_6BhQ · 2023-12-06

**Metareview:**

This paper studies the implicit bias of a continuous dynamic corresponding to a first-order approximation of the Momentum method for diagonal linear neural networks.
The authors show that the limit point of the continuous dynamics is less affected by the initialization. Also, they show that the gradient of the loss at the initialization affects the limit point of the dynamics.

One main concern is the connection between the continuous version considered and the practical discrete-time heavyball method. In particular, it seems that if you want the limit point of the continuous dynamics to correspond to one of the discrete dynamics, you need a vanishingly small step size (which trivializes the continuous formulation of HB to the continuous gradient dynamics).

**Justification For Why Not Higher Score:**

See my additionnal comments

**Justification For Why Not Lower Score:**

N/A

---

### Decision · Program_Chairs · 2024-01-16

Reject